# Convergence of inputs from the basal ganglia with layer 5 of motor cortex and cerebellum in mouse motor thalamus

Kevin P Koster, S Murray Sherman*

Department of Neurobiology, University of Chicago, Chicago, United States

**Abstract** A key to motor control is the motor thalamus, where several inputs converge. One excitatory input originates from layer 5 of primary motor cortex ($M1_{L5}$), while another arises from the deep cerebellar nuclei (Cb). $M1_{L5}$ terminals distribute throughout the motor thalamus and overlap with GABAergic inputs from the basal ganglia output nuclei, the internal segment of the globus pallidus (GPi), and substantia nigra pars reticulata (SNr). In contrast, it is thought that Cb and basal ganglia inputs are segregated. Therefore, we hypothesized that one potential function of the GABAergic inputs from basal ganglia is to selectively inhibit, or gate, excitatory signals from $M1_{L5}$ in the motor thalamus. Here, we tested this possibility and determined the circuit organization of mouse (both sexes) motor thalamus using an optogenetic strategy in acute slices. First, we demonstrated the presence of a feedforward transthalamic pathway from $M1_{L5}$ through motor thalamus. Importantly, we discovered that GABAergic inputs from the GPi and SNr converge onto single motor thalamic cells with excitatory synapses from $M1_{L5}$. Separately, we also demonstrate that, perhaps unexpectedly, GABAergic GPi and SNr inputs converge with those from the Cb. We interpret these results to indicate that a role of the basal ganglia is to gate the thalamic transmission of $M1_{L5}$ and Cb information to cortex.

*For correspondence:
msherman@bsd.uchicago.edu

Competing interest: The authors declare that no competing interests exist.

## Editor's evaluation

This study provides fundamental findings that challenge the traditional view of segregated cerebellar and basal ganglia circuits in the motor thalamus, revealing a novel intersection where GABAergic inputs from the basal ganglia converge with excitatory inputs from both the primary motor cortex and deep cerebellar nuclei. The evidence supporting these claims is compelling, utilizing rigorous optogenetic approaches and tricolor viral labeling to uncover the intricate circuit organization. These findings have theoretical and practical implications beyond a single subfield, advancing our understanding of how the brain processes motor commands.

## Introduction

The cerebral cortex, cerebellum, and basal ganglia work together to control motor behavior. Projections from these areas converge in the motor thalamus, a conglomerate of the ventral anterior (VA) and ventral lateral (VL) nuclei, often grouped together in rodents as VA/VL, as well as the ventral medial (VM) nucleus.

Cerebellar afferents to the motor thalamus appear to provide a driving excitatory input (**Gornati et al., 2018**), particularly to the region corresponding to VL (and VPL) in primates that, in turn, preferentially innervates primary motor cortex (M1) (**Sommer, 2003**; **Bosch-Bouju et al., 2013**). The VA and VL motor thalamic nuclei, which preferentially project to motor preparatory regions of cortex (**Sommer, 2003**; **Bosch-Bouju et al., 2013**), receive inputs from layer 5 (L5) of motor and frontal cortex

(*Rouiller et al., 1998*; *Rouiller et al., 2003*; *Kakei et al., 2001*; *Kultas-Ilinsky et al., 2003*; *Prasad et al., 2020*), although the synaptic properties of these inputs are unknown.

In sensory thalamic nuclei, L5 inputs initiate disynaptic circuits through the thalamus, called transthalamic pathways, that connect cortical regions indirectly (*Theyel et al., 2010*; *Blot et al., 2021*; *Miller-Hansen and Sherman, 2022*). These transthalamic pathways appear to be organized in parallel with direct corticocortical connections (*Theyel et al., 2010*; *Sherman and Guillery, 2011*; *Sherman and Guillery, 2013*; *Sherman, 2016*) and share a common organization across sensory modalities (*Miller-Hansen and Sherman, 2022*). However, it remains unclear whether transthalamic pathways also operate in the motor system.

There are two main GABAergic basal ganglia output nuclei that project to VA/VL: the internal segment of the globus pallidus (GPi) and the substantia nigra pars reticulata (SNr) (*Sommer, 2003*; *Lanciego et al., 2012*). The GPi is a main output of the BG to thalamus found in mammals, but its homolog in rodents is typically called the "entopeduncular nucleus." To avoid confusion, our use of "GPi" refers to this, even though we work in mice. Given these GABAergic inputs to thalamus, an intriguing possibility is that the basal ganglia intersect excitatory pathways through motor thalamus, providing a substrate by which the basal ganglia can gate information flow to cortex. However, a prevalent notion is that cerebellar and basal ganglia inputs are functionally segregated in the motor thalamus (*Bosch-Bouju et al., 2013*). Moreover, the traditional view of the basal ganglia projection to motor thalamus is that it is one step within the relatively closed cortical-basal ganglia-thalamocortical loop. This notion suggests that direct L5 inputs to motor thalamus might innervate a subpopulation of motor thalamic relays that do not participate in this classical loop.

There are two main GABAergic basal ganglia output nuclei that project to VA/VL: the internal segment of the globus pallidus (GPi) and the substantia nigra pars reticulata (SNr) (*Sommer, 2003*; *Lanciego et al., 2012*). The GPi is a main output of the BG to thalamus found in mammals, but its homolog in rodents is typically called the "entopeduncular nucleus." To avoid confusion, our use of "GPi" refers to this, even though we work in mice. Given these GABAergic inputs to thalamus, an intriguing possibility is that the basal ganglia intersect excitatory pathways through motor thalamus, providing a substrate by which the basal ganglia can gate information flow to cortex. However, a prevalent notion is that cerebellar and basal ganglia inputs are functionally segregated in the motor thalamus (*Bosch-Bouju et al., 2013*). Moreover, the traditional view of the basal ganglia projection to motor thalamus is that it is one step within the relatively closed cortical-basal ganglia-thalamocortical loop. This notion suggests that direct L5 inputs to motor thalamus might innervate a subpopulation of motor thalamic relays that do not participate in this classical loop.

Here, we addressed these three outstanding questions. First, we confirm the presence of a feedforward transthalamic pathway through the VA/VL segment of mouse motor thalamus. Second, we employed a dual opsin approach and determined that inhibitory afferents from the GPi and SNr to motor thalamus frequently converge on cells receiving excitatory driving inputs from L5 of motor cortex, demonstrating the transthalamic pathway may be gated by the basal ganglia. Finally, we also find functional convergence of inputs to VA/VL from the basal ganglia and cerebellar nuclei (Cb), suggesting these circuits are not wholly segregated. Taken together, our data highlight a role for the basal ganglia in gating the relay of driving excitatory inputs through motor thalamus, determining which information streams reach cortex at any one time.

## Results

### A feedforward transthalamic pathway through mouse VA/VL (Figure 1)

Higher order sensory thalamic nuclei, like the posterior medial nucleus or pulvinar (also known as the lateral posterior nucleus in rodents), receive robust inputs from cortical L5 and a fraction of these constitute feedforward or feedback transthalamic pathways (*Theyel et al., 2010*; *Mo and Sherman, 2019*; *Blot et al., 2021*; *Miller-Hansen and Sherman, 2022*). Physiological evidence in VM nucleus of the motor thalamic complex indicates cortico-thalamo-cortical interactions are mainly organized in reciprocal loops, demonstrating only a very limited feedforward transthalamic component from primary to secondary (i.e. anterolateral) motor cortex (*Guo et al., 2018*). Therefore, we used optogenetics to investigate whether a feedforward transthalamic pathway existed in the VA/VL portion of the motor thalamus; i.e., whether excitatory inputs from L5 of primary motor cortex (M1$_{L5}$) impinge on VA/

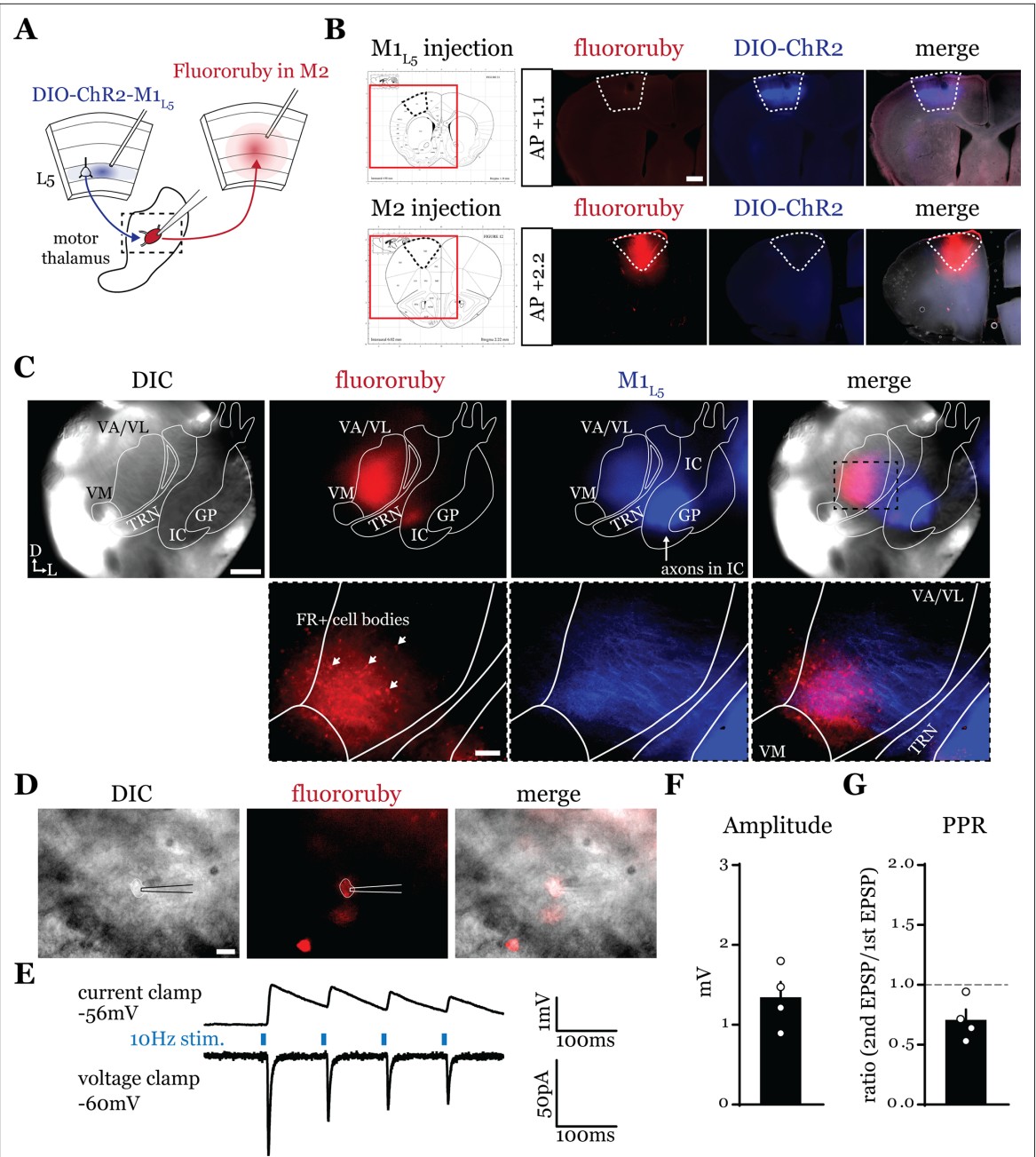

**Figure 1.** A feedforward transthalamic pathway through ventral anterior (VA)/ventral lateral (VL) motor thalamus. (**A**) Schematic of injections and experiment. (**B**) Representative of primary motor cortex (M1) (top row) and secondary motor cortex (M2) (bottom row) injection sites with the corresponding *Paxinos and Franklin, 2007*, atlas image, demonstrating no bleed through of one injection to the next. Scale = 500 μm. (**C**) Representative low-magnification (×4) differential interference contrast (DIC) and fluorescence images from the recording rig (top row) and post hoc histology of layer 5 of primary motor cortex (M1$_{L5}$) (ChR2) and M2 (fluororuby) labeling (bottom row) demonstrating a region of overlap between the two labels in the ventral VA/VL. Scale = 500 μm for top row, 100 μm for bottom row. (**D**) Representative high-magnification (×40 immersion) images from the recording rig demonstrating the recording electrode patched onto a fluororuby-positive neuron in VA/VL (same neuron the traces in E are derived from). Scale = 20 μm. (**E**) Representative traces in current (top) or voltage (bottom) clamp (held at –60 mV) of laser-evoked responses (blue rectangles) in an M2-projecting VA/VL thalamic neuron. (**F**) Quantification of the amplitude of the first excitatory postsynaptic potential (EPSP) in a 10 Hz pulse train for all cells in the transthalamic pathway (n=4). (**G**) Quantification of the paired-pulse ratio (PPR; amplitude of second EPSP/first EPSP in a 10 Hz pulse train) for all cells in the transthalamic pathway (n=4). Data are mean ± SEM.

The online version of this article includes the following source data for figure 1:

**Source data 1.** Amplitude and PPR values for *Figure 1F and G*.

VL relays that project to secondary motor cortex (M2) (*Figure 1A*). Specifically, we injected fluororuby into all layers of M2, recorded from retrogradely labeled (i.e. M2-projecting) neurons in VA/VL, and tested each for an M1$_{L5}$ input driven by Cre-dependent expression of ChR2 in Rbp4-Cre mice, which expresses Cre recombinase in L5 of cortex (*Figure 1B and C*). We found that from 25 fluororuby-positive cells across four mice (*Figure 1D*), 4 cells demonstrated a clear photostimulation-dependent excitatory response (*Figure 1E–G*).

## Anterograde labeling reveals overlap of motor thalamus inputs in ventral VA/VL (Figure 2)

While it is known that the motor cortex innervates the motor thalamus (*Figure 1*; *Kita and Kita, 2012*; *Economo et al., 2018*; *Winnubst et al., 2019*; *Prasad et al., 2020*), previous anatomical studies demonstrate two zones with minimal overlap in the motor thalamus, one innervated by the basal ganglia, and the other, by the deep Cb nuclei (*Anderson and DeVito, 1987*; *Sakai et al., 1996*; *Kuramoto et al., 2011*). Therefore, we next sought to directly compare the distribution of projections from the GPi, the Cb, and M1$_{L5}$ in the same mice by employing three-color anterograde labeling. We delivered red, green, and blue fluorophores via viral injection into the Cb, GPi, and M1, respectively (*Figure 2—figure supplement 1A–C*). We used Rbp4-Cre mice, and the injection of the label into M1 was Cre-dependent, thus limiting the label to cortical L5 neurons. We then analyzed their terminal distribution of these three afferent inputs in the motor thalamus (*Figure 2A–C*).

To generate an input map of motor thalamus, we averaged the fluorescence arising from each input across three animals and overlayed them onto the coronal mouse atlas (*Paxinos and Franklin, 2007*) at multiple rostro-caudal planes (*Figure 2C*). This map corroborates previous findings using anterograde labeling in rodents in several ways. First, we find that the Cb afferents are robust throughout much of motor thalamus but particularly concentrated in the lateral portions of rostral VA/VL, as well as the medial and dorsomedial aspects of caudal VA/VL; Cb afferents also sparsely innervate the rostral VM. Further, we observe that the GPi projection to motor thalamus, at least in comparison to the Cb inputs, is more limited, concentrating in the middle of rostral VA/VL with a proscribed patch in the ventrolateral aspect of VA/VL. In contrast to both of these inputs, we find that the input from M1$_{L5}$ is diffuse throughout VA/VL, appearing as coursing axons in large swaths of the region, though terminals do concentrate in VM, ventral VA/VL, and the medial boundary of VA/VL. Rostral to the sections examined, there was little representation from any input, while in more caudal sections (beyond –1.1 AP), GPi terminals are virtually absent, M1$_{L5}$ terminals are found primarily in VM, and Cb terminals target ventral VPL and VL (*Figure 2—figure supplement 1D and E*).

Importantly, the simultaneous labeling of all three inputs in the same animal(s) reveals a degree of overlap between inputs from GPi and Cb, particularly in the rostral VA/VL and ventrolateral VA/VL in more caudal (approximately –1.0 mm AP from bregma) sections (*Figure 2C*). Terminal zones from M1$_{L5}$ and Cb overlapped in central and ventral aspects of the VA/VL, while overlap is also likely in the more dorsal aspects where M1$_{L5}$ axons diffusely project (*Figure 2C*). Generally, Cb inputs overlapped with those from GPi laterally and dorsally, while M1$_{L5}$ inputs tended to overlap with GPi medially. A quantitative analysis of the convergence between each set of inputs demonstrates that approximately 10% of the M1$_{L5}$ and GPi input zones are overlapping (*Figure 2D*). Similarly, between 10% and 20% of the Cb and GPi terminals zones are overlapping between –0.8 and –1.0 AP (*Figure 2D*).

## GABAergic inputs from GPi converge with glutamatergic inputs from both M1$_{L5}$ and Cb (Figure 3)

Having defined the regions of overlap between GPi and M1$_{L5}$, we next tested whether these inputs converge onto single thalamic cells using acute slice physiology in Rbp4 mice (*Figure 3A*). Our data for these experiments include 62 cells from 14 animals, though we focus on cells receiving at least one input (42/62 cells). We transduced GPi neurons with ChR2-GFP (or, in some animals, Dlx-ChR2 for inhibitory neuron selective expression) and transduced M1$_{L5}$ neurons with cre-dependent ChR2-mCherry (*Figure 3A and B*). Since GPi outputs are GABAergic and M1$_{L5}$, glutamatergic, we readily identified their evoked activity on postsynaptic neurons recorded in the motor thalamus using a low-chloride internal pipette solution (5 mM Cl⁻ total) and clamping the cell at distinct holding potentials. Specifically, we recorded at a holding potential of –60 mV to accentuate excitatory postsynaptic currents (EPSCs) from M1$_{L5}$ and at –40 mV for inhibitory postsynaptic currents (IPSCs) from GPi (*Figure 3C*).

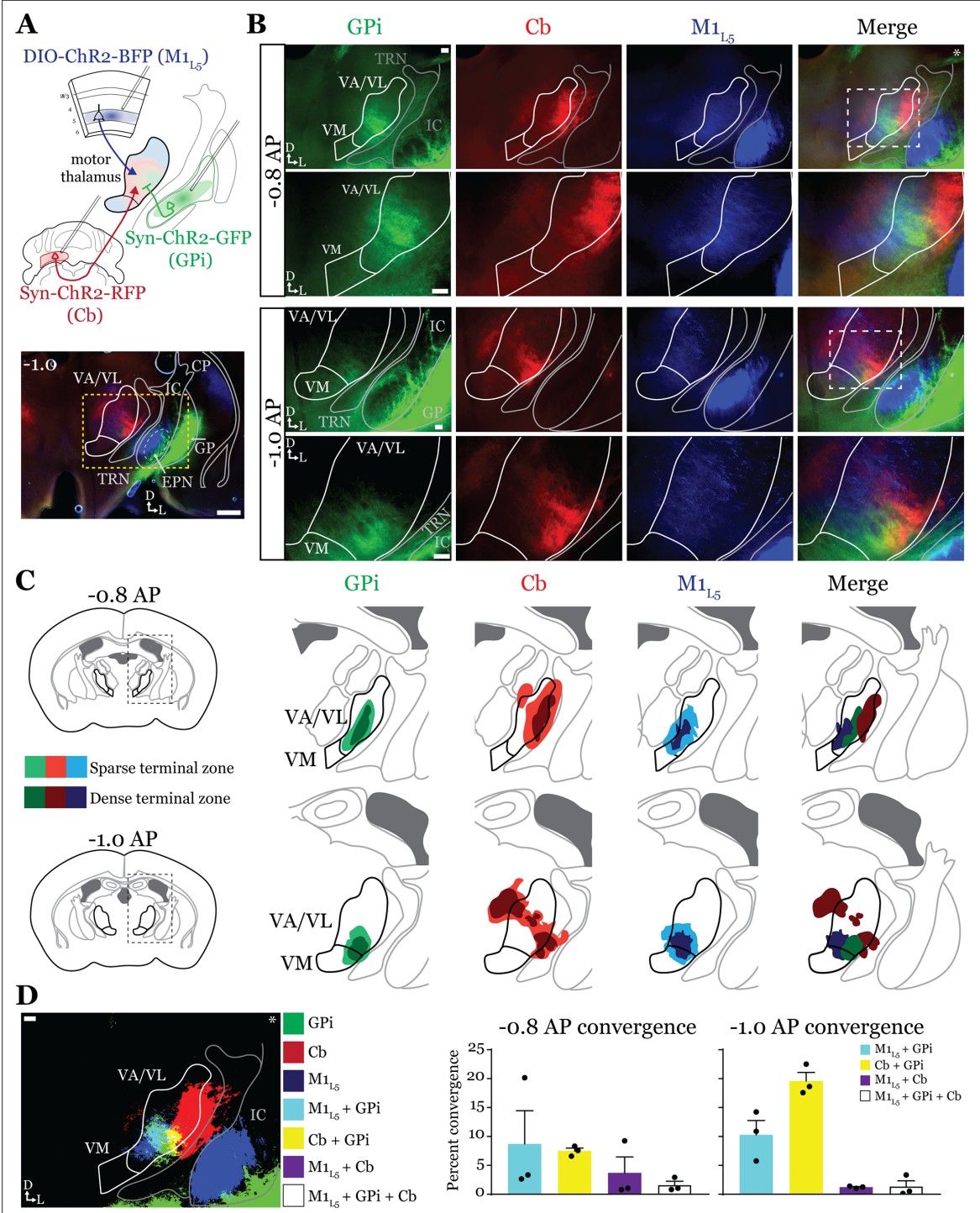

**Figure 2.** Anterograde labeling reveals overlap of inputs in ventral anterior (VA)/ventral lateral (VL) motor thalamus. (**A**) Schematic of injections and experiment (top) with representative low-magnification image (bottom) of a section demonstrating the three terminal zones in motor thalamus and internal segment of the globus pallidus (GPi) injection site. White dashed line represents the entopeduncular nucleus (EPN), referred to here as the GPi, which shows expression of virus in addition to the globus pallidus (GP) lateral to the internal capsule (IC). Yellow dashed line represents the area of higher magnification images shown in B. Scale = 500 μm. (**B**) Representative images of coronal sections at –0.8 mm and –1.0 mm AP from bregma. Injection site and terminals from GPi (green), terminals from cerebellar nuclei (Cb) (red), and layer 5 of primary motor cortex (M1$_{L5}$) axons (blue) coursing through the internal capsule and innervating motor thalamus are visible. White dashed line represents the area imaged in higher magnification in the next row of images. Asterisk demarks the same image used for convergence analysis in D. Scale = 100 μm for all images. (**C**) Cross-animal (n=3) averaged input maps of motor thalamus inputs at –0.8 mm and –1.0 mm AP from bregma color coded as in A and B. Dark colors represent dense

*Figure 2 continued on next page*

*Figure 2 continued*

terminal fields, while lighter colors represent sparser terminal fields. In the merged image (right), only outlines of the dense terminal fields are included. (**D**) Representative image depicting the terminal fields from each input with bit masks color coded as in A–C and quantification (right) of terminal field overlap (see Materials and methods) for sections at –0.8 mm and –1.0 mm AP from bregma. See *Figure 2—figure supplement 1* for representative injection sites and additional images of the rostral-caudal extent of inputs to motor thalamus.

The online version of this article includes the following source data and figure supplement(s) for figure 2:

**Source data 1.** Convergence values for *Figure 2D*.

**Figure supplement 1.** The examined inputs are segregated at the rostral and caudal poles of the motor thalamus.

---

To confirm that the inputs were in fact GABAergic or glutamatergic in a subset of recorded cells, we pharmacologically inhibited GABA$_A$ receptors or ionotropic glutamate receptors (iGluRs), respectively. In cells used for representative traces, bath application of gabazine was followed by a washout period to allow recovery of the GABA$_A$ response, after which DNQX was applied. In motor thalamic relays receiving only one input, blocking the associated receptor subtype completely abolished the evoked response at all holding potentials (*Figure 3C*, see GPi or M1$_{L5}$ only representative cells), while in relay cells receiving both inputs, the non-antagonized receptor system remained functional (*Figure 3C*, see 'both inputs' representative cell).

For each of the 42 cells studied, we analyzed if it received input from only GPi, only M1$_{L5}$, or received convergent input from both sources (*Figure 3D*). Whereas a majority (32/42; 76.2%) of cells received only one input, either from GPi (22/42; 52.4%) or M1$_{L5}$ (10/42; 23.8%), a minority of cells (10/42; 23.8%) received convergent input from both regions, evidenced by a mixed EPSC/IPSC response to laser stimuli (*Figure 3C and D*). Cataloging the location of all recorded cells revealed that motor thalamic neurons receiving inputs from both GPi and M1$_{L5}$ were spread throughout the rostral VA/VL but clustered relatively tightly around the distinct GPi terminal patch in more caudal sections (*Figure 3E*), as expected from our anatomical data (*Figure 2*).

The zones of motor thalamus innervated by the GPi and Cb are thought to be largely, if not entirely, independent (*Bosch-Bouju et al., 2013*; *Nakamura et al., 2014*). Therefore, we next tested whether excitatory inputs from the Cb also converge with those from the GPi (*Figure 3F and G*). Our data for these experiments includes 50 cells from eight animals, though, again, we analyze only those receiving an input (43 cells).

Activations of Cb inputs evoked large EPSCs, often producing an action potential or a burst of action potentials, mirroring a recent report (*Schäfer et al., 2021*). Like the pattern of GPi and M1$_{L5}$ inputs, a substantial proportion of cells (27/43; 62.8%) tested received only GPi (8/43; 18.6%) or only Cb input (19/43; 44.2%). Intriguingly, a sizable proportion received convergent input from both sources (16/43; 37.2%) (*Figure 3H and I*). Spatial analysis of all recorded cells showed a similar pattern to neurons receiving convergent input from GPi and M1$_{L5}$—neurons innervated by GPi and Cb were primarily found in the central and ventrolateral aspects of the VA/VL (*Figure 3J*).

## GABAergic inputs from SNr also converge with both M1$_{L5}$ and Cb (Figures 4 and 5)

Next, we tested whether the same organization applied to inputs from the second major output of the basal ganglia direct pathway, the SNr. We performed the same tricolor labeling experiment as above, substituting the GPi for the SNr (*Figure 4A* and *Figure 4—figure supplement 1A–C*). In general, terminations from the SNr showed a similar organization to those from the GPi, with the important exception that the SNr terminal zone was shifted medially and ventrally compared to the GPi, covering the ventromedial VA/VL and VM in the sections examined (*Figure 4B and C*). Therefore, SNr terminals demonstrated little overlap with those from the cerebellum but exhibited substantial overlap with M1$_{L5}$ terminals (*Figure 4D*).

Accordingly, analysis of 19 cells (35 cells total recorded, 16 with no detectable inputs) from five mice in the overlap zone between M1$_{L5}$ and SNr terminals (*Figure 5A and B*) demonstrated that the SNr indeed converges with inputs from M1$_{L5}$ on single cells (5/19 cells, 26.3%) (*Figure 5C and D*). Of the cells receiving an excitatory input (M1$_{L5}$), nearly half also had convergent input from SNr (45.5%) (*Figure 5D*).

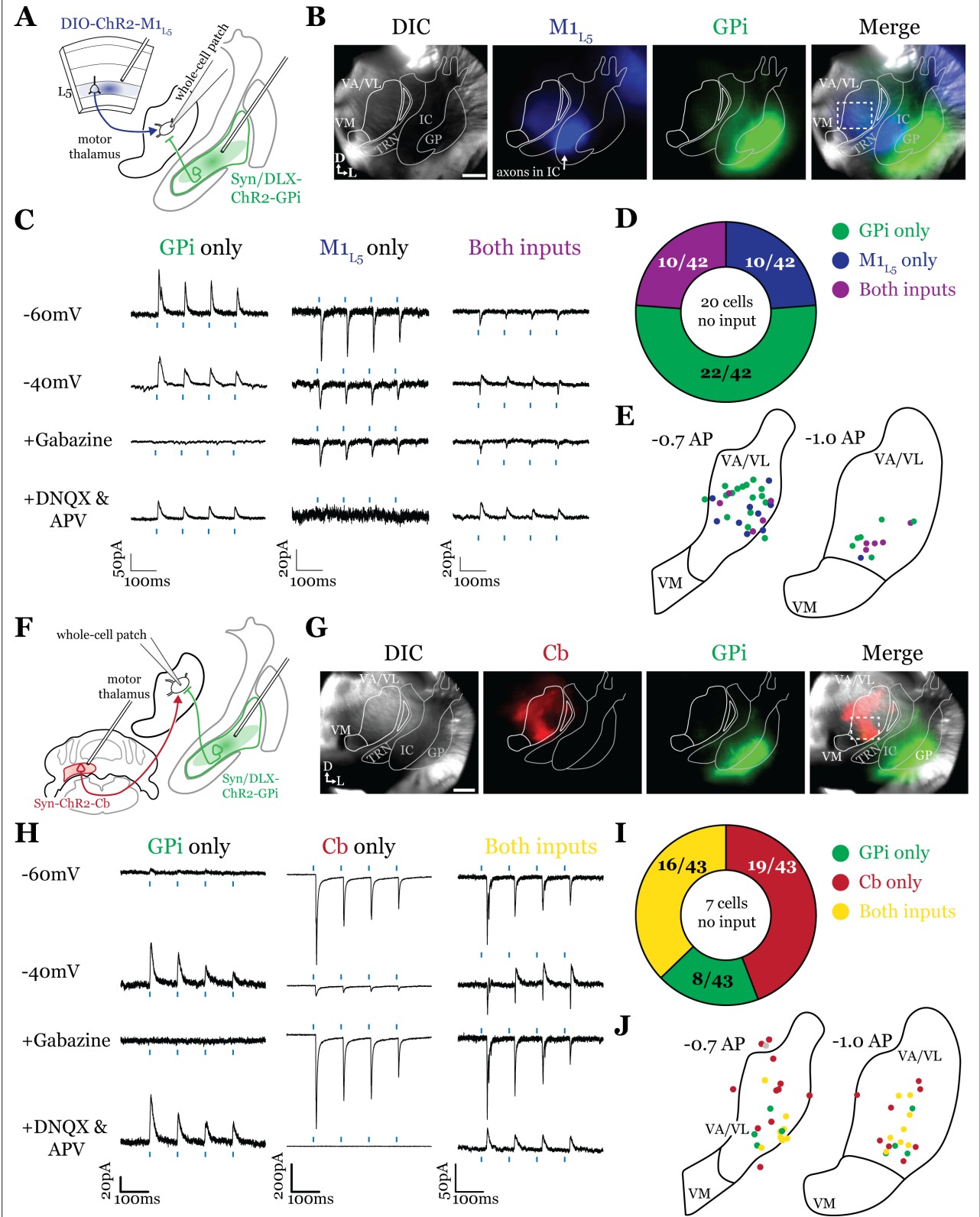

**Figure 3.** Internal segment of the globus pallidus (GPi) inputs to motor thalamus converge with those from layer 5 of primary motor cortex (M1$_{L5}$) and cerebellar nuclei (Cb). (**A**) Schematic of experiment. (**B**) Representative images from recording apparatus of the acute coronal slice (left) showing injection sites into GPi and terminals in motor thalamus (green), M1$_{L5}$ axons coursing through the internal capsule and terminals in motor thalamus (blue), and merged image. Dashed rectangle indicates approximate target of recording. Arrows in bottom left of image represent dorsal (D) and lateral

*Figure 3 continued on next page*

*Figure 3 continued*

(L) directions. Scale = 0.5 mm. (**C**) Representative traces in voltage clamp of laser-evoked responses (blue rectangles) in three motor thalamic neurons (recorded with low Cl- internal solution) held near the resting membrane potential (Vm, –60), at –40 mV, at –40 mV with application of the GABA$_A$ antagonist gabazine (20 µM), and at –60 mV with application of the ionotropic glutamate receptor (iGluR) antagonists, DNQX (50 µM) and AP5 (100 µM). Note that some cells only exhibit inhibitory postsynaptic currents (IPSCs), some only excitatory postsynaptic currents (EPSCs), and some demonstrate a mixed EPSC/IPSC response, of which the excitatory or inhibitory components are abolished by iGluR or GABA$_A$ blockade, respectively. For quantification of the synaptic properties of excitatory inputs, see *Figure 6D*. (**D**) Summary data depicting how many recorded cells receive only GPi input, only M1$_{L5}$ input, both, or neither. (**E**) Spatial pattern of motor thalamus neurons receiving each input type color coded to the inputs they receive. (**F**) Schematic of experiment. (**G**) Representative images from recording apparatus of the acute coronal slice (left) showing injection sites into GPi and terminals in motor thalamus (green), Cb terminals in motor thalamus (red), and merged image. Dashed rectangle indicates approximate target of recording. Arrows in bottom left of image represent dorsal (D) and lateral (L) directions. Scale = 0.5 mm. (**H**) Representative traces in voltage clamp of laser-evoked responses (blue rectangles) in three motor thalamic neurons (recorded with low Cl- internal solution) held near the resting membrane potential (Vm, –60), at –40 mV, at –40 mV with application of the GABA$_A$ antagonist gabazine (20 µM), and at –60 mV with application of the iGluR antagonists, DNQX (50 µM) and AP5 (100 µM). Note that some cells only exhibit IPSCs, some only EPSCs, and some demonstrate a mixed EPSC/IPSC response, of which the excitatory or inhibitory components are abolished by iGluR or GABA$_A$ blockade, respectively. (**I**) Summary data depicting how many recorded cells receive only GPi input, only Cb input, both, or neither. (**J**) Spatial pattern of motor thalamus relays color coded to the inputs they receive.

Surprisingly, analysis of 19 cells (30 cells recorded in total, 11 with no detectable inputs) from seven mice demonstrates that SNr inputs also converged with excitatory projections from the Cb (6/19 cells, 31.6%) (*Figure 5F–J*). This suggests that while the anatomical overlap of dense terminal fields between SNr and Cb is quite limited (dark colored areas in *Figure 4C*; *Figure 4D*), the areas of sparser innervation are sufficient to find a substantial proportion of cells receiving both inputs.

## Both Cb and M1$_{L5}$ provide driver innervation of motor thalamus (Figure 6)

Glutamatergic synapses can be broadly classified into two types, drivers and modulators (*Sherman and Guillery, 1998*; *Sherman, 2016*). Driver-type synapses are considered the primary source of information for the postsynaptic cell, while modulator-type synapses modify how that information is received (*Sherman, 2016*). Anatomical analysis cannot definitively distinguish between drivers and modulators, because while all modulator synapses are relatively small (<1 µm$^2$) and drivers are generally large (>2 µm$^2$), driver synapses span the entire distribution of terminal sizes (*Petrof and Sherman, 2013*; *Prasad et al., 2020*), being both small and large with the potential for subtle functional differences between subclasses (*Viaene et al., 2011*; *Koster and Carroll, 2022*). However, drivers and modulators are clearly distinguished physiologically, as drivers exhibit a larger initial excitatory postsynaptic potential (EPSP), paired-pulse depression, and no activation of metabotropic glutamate receptors (mGluRs). In contrast, modulators display paired-pulse facilitation, smaller initial EPSPs, and activate mGluRs.

Conflicting evidence exists for the characteristics of M1$_{L5}$ and Cb terminals in motor thalamus, such that the nature of their synaptic properties (driver vs. modulator) were ambiguous (*Deniau et al., 1992*; *Rouiller et al., 1998*; *Rouiller et al., 2003*; *Kakei et al., 2001*; *Kultas-Ilinsky et al., 2003*; *Kuramoto et al., 2011*; *Rovó et al., 2012*). These studies led us to explicitly investigate the synaptic properties of both excitatory pathways to motor thalamus.

To obtain reliable paired-pulse effects, we performed laser stimulation of ChR2-positive axons >300 µm from the recorded cell rather than at ChR2-positive terminals, since terminal photostimulation produces unreliable paired-pulse effects (*Jackman et al., 2014*; *Mo and Sherman, 2019*). We found that both M1$_{L5}$ and Cb have driver-type synaptic properties. Specifically, these inputs showed synaptic depression at both low (10 Hz) and high (40 Hz is shown) photostimulation frequencies, an insensitivity to antagonists of mGluRs, and responses that were completely abolished by blockade of iGluRs (*Figure 6A and B*). Note that although ChR2(H134R) does not perform optimally beyond 20 Hz, our previous data demonstrate that such optical high-frequency stimulation is sufficient to stimulate an mGluR response under these conditions (*Miller-Hansen and Sherman, 2022*), which we do not detect here. Compilation of the paired-pulse ratio (PPR) (second EPSC/first EPSC) for each cell tested showed a depressing phenotype for these excitatory inputs to motor thalamus (*Figure 6D*).

Although only glutamatergic synapses can be considered for classification in the driver/modulator scheme, GABAergic inputs share a parallel in that only some synapses activate metabotropic GABA$_B$ receptors (*Sherman, 2007*). These receptors are one potential source of the type of sustained

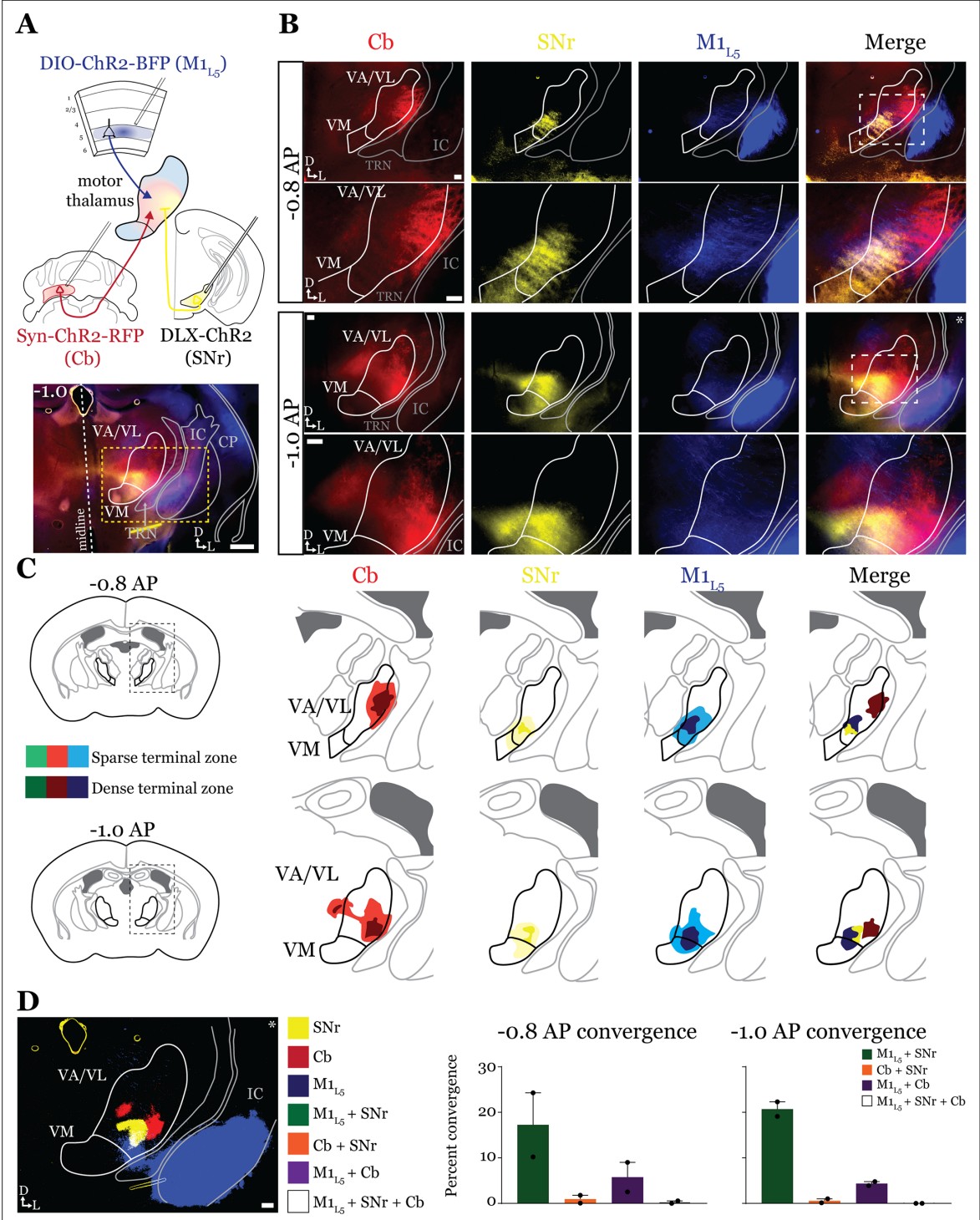

**Figure 4.** Anterograde labeling reveals stronger overlap between substantia nigra pars reticulata (SNr) and layer 5 of primary motor cortex (M1$_{L5}$) compared to cerebellar nuclei (Cb) and SNr in ventral anterior (VA)/ventral lateral (VL) motor thalamus. (**A**) Schematic of injections and experiment (top) with representative low-magnification image (bottom) of a section demonstrating the three terminal zones in motor thalamus. Yellow dashed line represents the area of higher magnification images shown in B. Scale = 500 μm. (**B**) Representative images of coronal sections at –0.8 mm and –1.0 mm AP from bregma. Terminals from SNr (yellow), terminals from Cb (red), and M1$_{L5}$ axons (blue) coursing through the internal capsule and innervating motor thalamus are visible. White dashed line represents the area imaged in higher magnification in the next row of images. Asterisk demarks the same image represented for convergence analysis in D. Scale = 100 μm for all images. (**C**) Cross-animal (n=2) averaged input maps of motor thalamus inputs at –0.8 mm and –1.0 mm AP from bregma color coded as in A and B. Dark colors represent dense terminal fields, while lighter colors represent sparser terminal fields. In the merged image (right), only outlines of the dense terminal fields are included. (**D**) Representative image depicting the terminal

*Figure 4 continued on next page*

*Figure 4 continued*

fields from each input with bit masks color coded as in A–C and quantification (right) of terminal field overlap (see Materials and methods) for sections at –0.8 mm and –1.0 mm AP from bregma. See *Figure 4—figure supplement 1* for representative injection sites.

The online version of this article includes the following source data and figure supplement(s) for figure 4:

**Source data 1.** Convergence values for *Figure 4D*.

**Figure supplement 1.** Representative injection sites for tricolor-labeled substantia nigra pars reticulata (SNr) animal.

hyperpolarization required for the characteristic rebound spiking of thalamic relay neurons (*Llinás and Jahnsen, 1982*), and may therefore play an important role in the gating of motor thalamocortical communication. However, it is unknown whether the inhibitory input from the GPi activates metabotropic GABA_B receptors. Therefore, we tested the synaptic properties of the GPi input to motor thalamus and, particularly, whether we could detect the presence of GABA_B receptor activation. The GPi input to all cells tested was depressing at both low (10 Hz) and high (40 Hz) stimulation frequencies, unchanged by GABA_B blockade, and completely abolished by gabazine bath application (*Figure 6C*). Therefore, we conclude that at least in the population of cells tested, GPi inputs to motor thalamus do not activate GABA_B receptors.

## Cb terminals within or outside the GPi overlap zone are not different in size (Figure 7)

Next, we directly tested the qualitative finding (*Deniau et al., 1992*) that Cb terminals in the basal ganglia overlap zones are smaller than those in the non-overlapping zone. We used dual anterograde labeling and performed terminal size analysis of the region in ventrolateral VA/VL where Cb and GPi overlap, versus the area immediately surrounding the overlap zone (*Figure 7*). Both Cb terminal size distribution (*Figure 7B*) and average size (*Figure 7C*) were the same between the overlap ('within GPi zone') and non-overlapping ('outside GPi zone') regions (within = 2.53 ± 0.1 vs. outside = 3.03 ± 0.3 $\mu m^2$; p=0.222 by Mann-Whitney $U$-test). However, the number of terminals in a size-matched region outside the GPi overlap zone was greater (within = 223.1 ± 37.4 vs. outside = 458.8 ± 88.0 terminals; *p=0.0317 by Mann-Whitney $U$-test) (*Figure 7D*), supporting the notion that Cb terminals invading the basal ganglia overlap zones are sparser (*Deniau et al., 1992*). Nevertheless, the finding that terminals outside the GPi overlap zone are no different in size to those within it agree with the functional evidence herein (i.e. *Figure 5*) and elsewhere (*Gornati et al., 2018*) that cerebellar inputs to motor thalamus are uniformly drivers. It is plausible that Cb terminals in more caudal areas of the motor thalamus where Cb innervates VM are indeed smaller in size (*Deniau et al., 1992*), but we focused here on VA/VL where we made our recordings.

## Discussion

Our data provide the first functional evidence for a feedforward transthalamic pathway through the mouse VA/VL. Further, we show that both GPi and SNr efferents converge with those from M1_{L5} or Cb onto single motor thalamic neurons. Note that this is quite different from previous demonstrations of convergence of Cb with layer 6 (L6) of cortex (*Schäfer et al., 2021*), as it is anticipated that every, or nearly every, thalamic relay receives a modulatory L6 input, whereas L5 projections are more selective and are drivers (*Sherman and Guillery, 2013*; *Usrey and Sherman, 2019*). Furthermore, whereas we show anatomically that there are regions in motor thalamus in which terminals from M1_{L5}, Cb, and basal ganglia overlap, raising the possibility of triple convergence onto thalamic neurons there, we have not demonstrated triple convergence functionally. This remains a possibility for future study.

### Experimental provisos

Whereas our data reveal that the GPi and SNr can converge with both the M1_{L5} and Cb pathways to motor thalamus, only a subset of thalamic neurons receives such convergent input. There are several potential reasons for the seeming sparsity of cells receiving the inputs of interest, particularly regarding M1_{L5}. Several frontal and motor cortical areas project to the motor thalamus, concentrating in distinct subsections (*Bosch-Bouju et al., 2013*). Therefore, by focusing our attention (i.e. injections) on M1, we substantially limited the number of thalamic relays potentially receiving an L5 input. Moreover, we

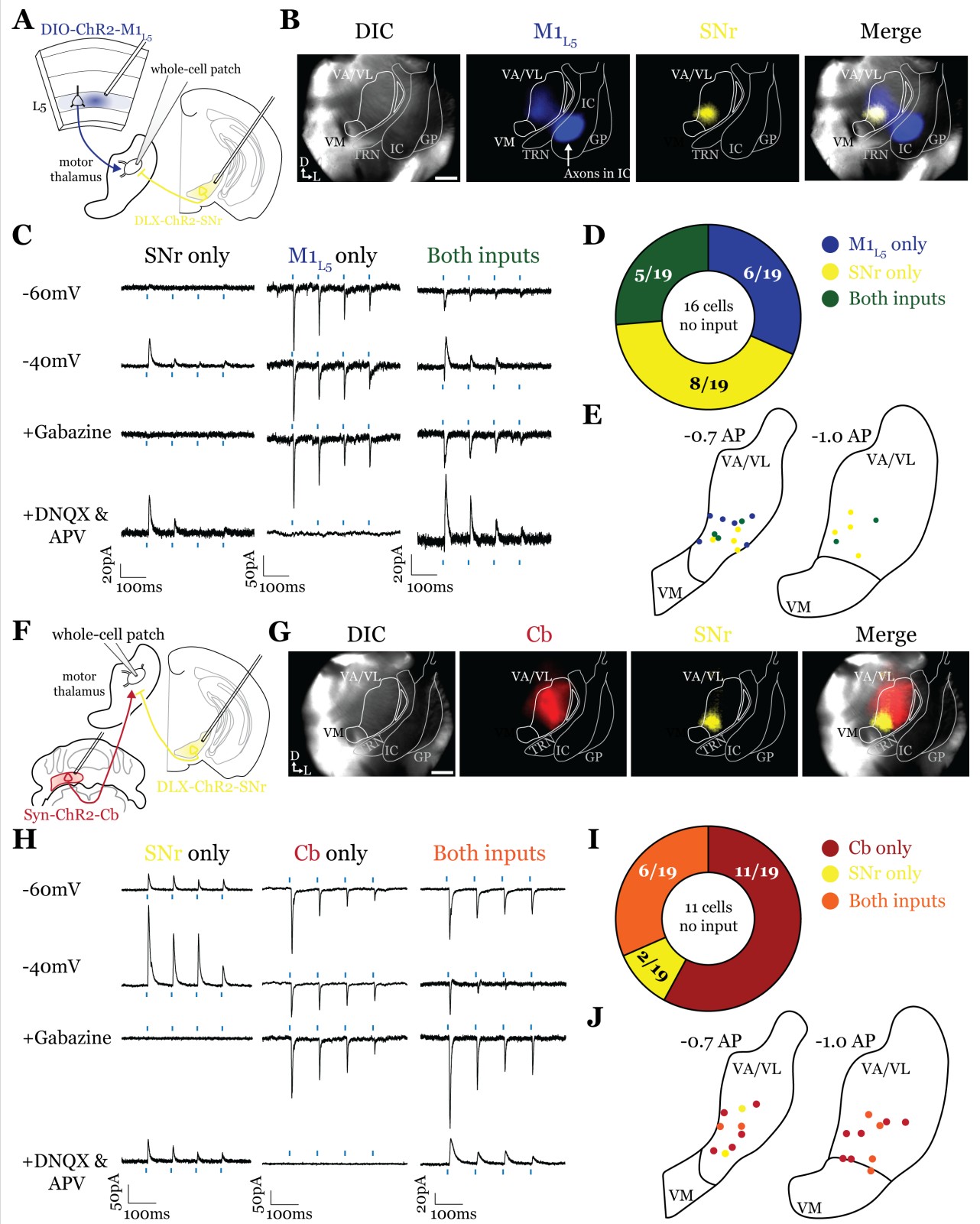

**Figure 5.** Substantia nigra pars reticulata (SNr) inputs to motor thalamus converge with those from layer 5 of primary motor cortex (M1_L5) and cerebellar nuclei (Cb). (**A**) Schematic of experiment. (**B**) Representative images from recording apparatus of the acute coronal slice showing SNr synaptic terminals in the motor thalamus (yellow), and M1_L5 axons invading the motor thalamus through the internal capsule (blue), and merged image. Arrows in bottom left of image represent dorsal (D) and lateral (L) directions. Scale = 0.5 mm. (**C**) Representative traces in voltage clamp of laser-evoked responses

*Figure 5 continued on next page*

*Figure 5 continued*

(blue rectangles) in three motor thalamic neurons (recorded with low Cl- internal solution) held near the resting membrane potential (Vm, –60), at –40 mV, at –40 mV with application of the GABA$_A$ antagonist gabazine (20 μM), and at –60 mV with application of the ionotropic glutamate receptor (iGluR) antagonists, DNQX (50 μM) and AP5 (100 μM). Note that some cells only exhibit inhibitory postsynaptic currents (IPSCs), some only excitatory postsynaptic currents (EPSCs), and some demonstrate a mixed EPSC/IPSC response, of which the excitatory or inhibitory components are abolished by iGluR or GABA$_A$ blockade, respectively. For quantification of the synaptic properties of excitatory inputs, see *Figure 6D*. (**D**) Summary data depicting how many recorded cells receive only SNr input, only M1$_{L5}$ input, both, or neither. (**E**) Spatial pattern of motor thalamus neurons receiving each input type color coded to the inputs they receive. (**F**) Schematic of experiment. (**G**) Representative images from recording apparatus of the acute coronal slice (left) showing SNr terminals in motor thalamus (yellow), Cb terminals in motor thalamus (red), and merged image. Arrows in bottom left of image represent dorsal (**D**) and lateral (**L**) directions. Scale = 0.5 mm. (**H**) Representative traces in voltage clamp of laser-evoked responses (blue rectangles) in three motor thalamic neurons (recorded with low Cl- internal solution) held near the resting membrane potential (Vm, –60), at –40 mV, at –40 mV with application of the GABA$_A$ antagonist gabazine (20 μM), and at –60 mV with application of the iGluR antagonists, DNQX (50 μM) and AP5 (100 μM). Note that some cells only exhibit IPSCs, some only EPSCs, and some demonstrate a mixed EPSC/IPSC response, of which the excitatory or inhibitory components are abolished by iGluR or GABA$_A$ blockade, respectively. (**I**) Summary data depicting how many recorded cells receive only SNr input, only Cb input, both, or neither. (**J**) Spatial pattern of motor thalamus relays color coded to the inputs they receive.

targeted our recordings to the terminal overlap zones between any two sets of inputs. As revealed by our anatomical evidence (*Figures 2 and 4*), large proportions of motor thalamus are segregated to a single input (e.g. Cb versus SNr), again limiting the connection probability of recorded cells.

Within the overlap zone, however, the proportion of cells we find with convergent inputs likely reflects an underestimate of the actual numbers, because there are several additional reasons for false negatives; i.e., failure to detect an existing input. For instance, not all projection neurons from basal ganglia or cortex are likely to be transfected with ChR2, especially since many L5 neurons do not express Cre in the Rbp4 mouse line (*Harris et al., 2014*). Also, if topographic alignment is necessary to detect such convergence, it may not have been achieved in all cases. As mentioned above, the topography of cortical projections to motor thalamus may be particularly relevant. Given these caveats, absent a specific percentage, we conclude that a sizable, functionally relevant population of neurons within the overlap zones are gated. It should be noted that our conclusions below are focused on this subset of cells receiving convergent inputs.

## Both excitatory pathways to motor thalamus are drivers

The identification of driver inputs to thalamus is an important step in understanding the functioning of thalamic relays. Where identified, such inputs represent the main source of information to be relayed to cortex (e.g. retinal input to the LGN) (reviewed in *Sherman and Guillery, 1998*; *Sherman and Guillery, 2013*; *Usrey and Sherman, 2019*). Furthermore, identifying L5 as the driver input to many thalamic regions clarified the function of these transthalamic relays, which are now an accepted feature of cortical organization (*Sherman and Guillery, 1998*; *Sherman and Guillery, 2013*; *Usrey and Sherman, 2019*). In this study, we have used criteria from our recordings to identify the M1$_{L5}$ and Cb inputs to motor thalamus as driver.

While previous tracing studies in cats and monkeys (*Rouiller et al., 1998*; *Rouiller et al., 2003*; *Kakei et al., 2001*; *Kultas-Ilinsky et al., 2003*) showed that L5 of motor cortex projects with large terminals to the motor thalamus, which would indicate drivers, histological studies in monkeys and mice called into question whether large corticothalamic terminals existed in much of motor thalamus (*Kuramoto et al., 2011*; *Rovó et al., 2012*). We found here that the M1$_{L5}$ inputs to all motor thalamic cells displayed driver characteristics; however, EPSC amplitudes of ChR2-evoked responses from M1$_{L5}$ were indeed smaller than those from Cb inputs, which often elicited action potentials (see below). One potential reason for this difference is the degree of convergence among L5 corticothalamic inputs, compared to the first-order pathway from Cb, onto motor thalamic neurons. That is, whereas the total input from Cb or L5 might be equally strong, that from a single cortical area might represent only part of the driving input.

Prior anatomical data also showed Cb terminals innervating motor thalamic regions co-innervated by the basal ganglia were diffuse and small (*Deniau et al., 1992*), often characteristic of modulators (*Sherman, 2007*; *Sherman and Guillery, 2009*). However, we find that all Cb inputs to motor thalamic neurons are drivers, including a fraction of thalamic relays that exhibited smaller evoked EPSCs from Cb. These observations agree with a recent report characterizing Cb inputs across motor thalamic nuclei (*Gornati et al., 2018*), and are in line with the findings that driver synapses can occur

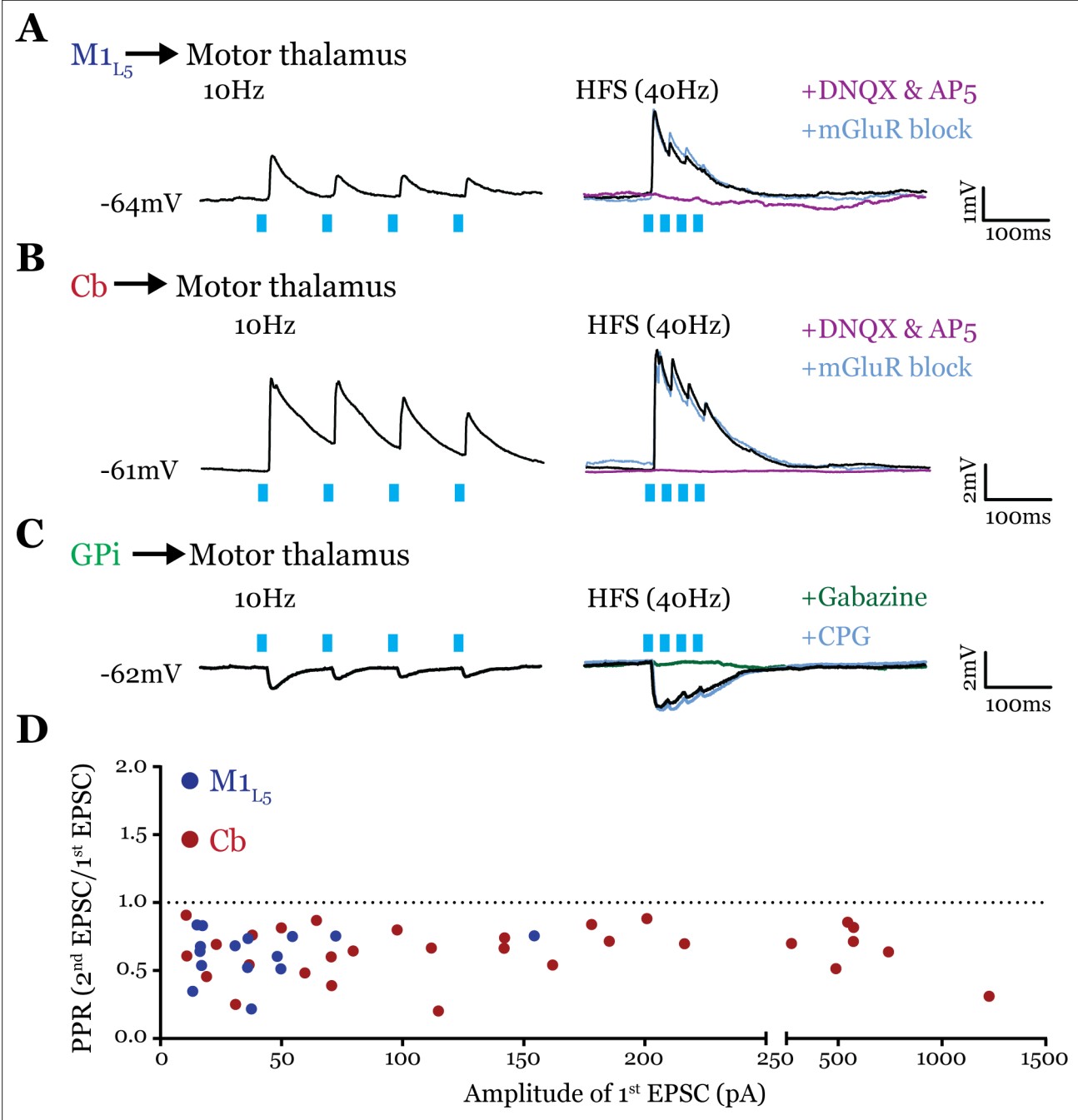

**Figure 6.** All inputs examined are drivers. (**A**) Representative current clamp recordings of motor thalamic neuron demonstrating distal activation of layer 5 of primary motor cortex (M1$_{L5}$) axons (>300 µm from recorded cell) gives a depressing response at low (10 Hz, left) and high frequencies (40 Hz, right) that is insensitive to metabotropic glutamate receptor (mGluR) antagonists (40 µM LY367385 and 30 µM MPEP) but abolished by DNQX. (**B**) Representative current clamp recordings of motor thalamic neuron demonstrating distal activation of cerebellar nuclei (Cb) axons (>300 µm from recorded cell) gives a depressing response at low (10 Hz, left) and high frequencies (40 Hz, right) that is insensitive to mGluR antagonists (40 µM LY367385 and 30 µM MPEP) but abolished by DNQX. (**C**) Representative current clamp recordings of motor thalamic neuron demonstrating distal activation of internal segment of the globus pallidus (GPi) axons (>300 µm from recorded cell) gives a depressing response at low (10 Hz, left) and high frequencies (40 Hz, right) that is insensitive to GABA$_B$ antagonists (CGP 46381, 25 µM) but abolished by gabazine (20 µM). (**D**) Compiled paired-pulse ratio (PPR) (second EPSC/first EPSC) data for all cells recorded receiving only the input of interest (not mixed inputs) plotted according to the amplitude of the first EPSC. PPRs below 1.0 (dotted line) are considered depressing, while those above 1.0 are considered facilitating. For Cb, n=30 cells; for M1$_{L5}$, n=15 cells. EPSC, excitatory postsynaptic current.

The online version of this article includes the following source data for figure 6:

**Source data 1.** EPSC amplitude and PPR values for all cells receiving an excitatory input as shown in *Figure 6D*.

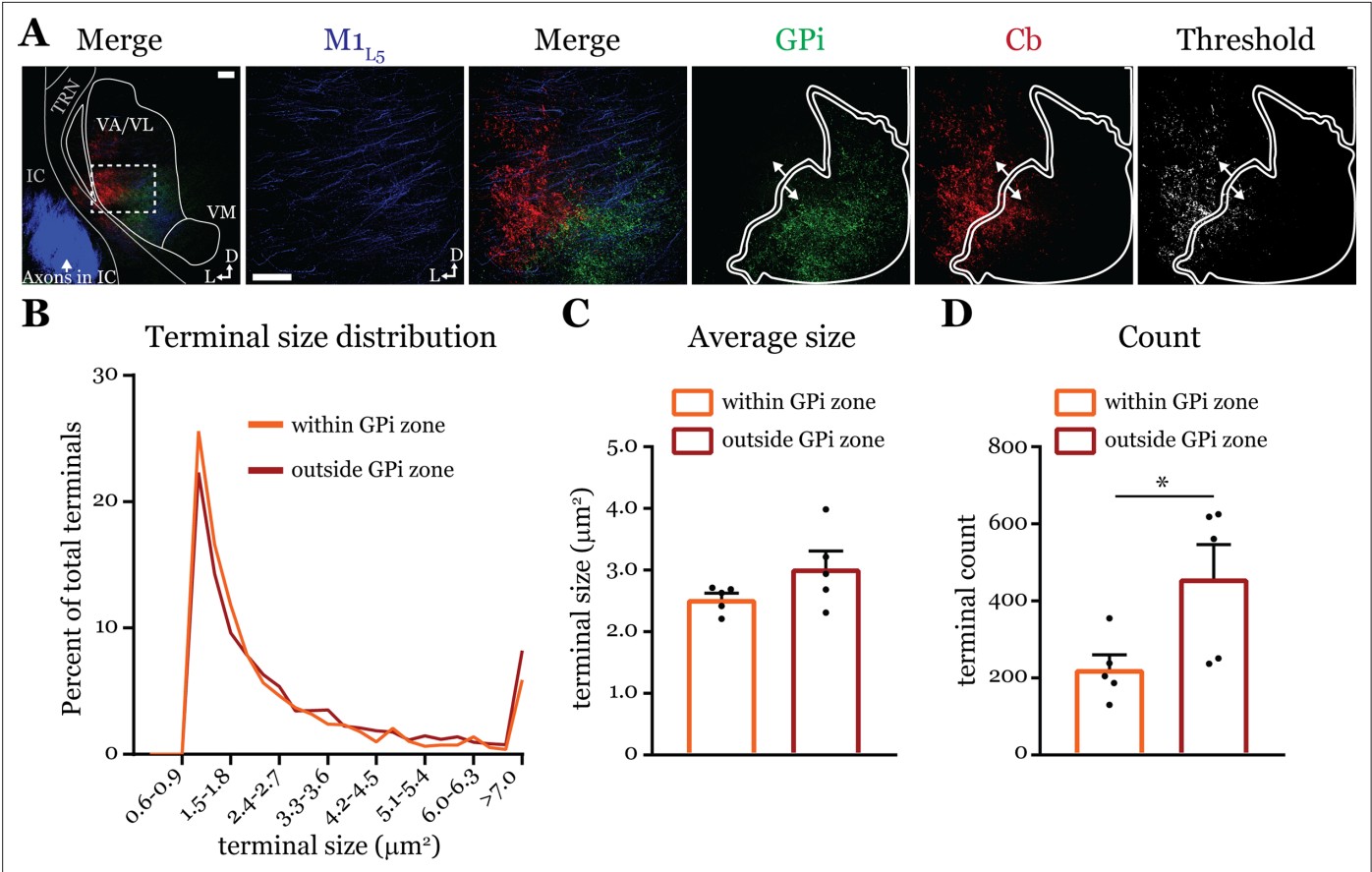

**Figure 7.** Cerebellar nuclei (Cb) terminal sizes within and outside the internal segment of the globus pallidus (GPi) overlap zone are equal. (**A**) Representative maximum intensity projections of terminal overlap fields in low magnification (*left*) and terminals from motor thalamic inputs with GPi and Cb terminal zones distinguished. Thresholded image demonstrates binary masking of terminals for high-throughput estimation of terminal size. Scale = 100 μm. (**B**) Population distribution of Cb terminal size averaged across five animals (two sections per animal, averaged). (**C**) Average Cb terminal sizes within and outside the GPi overlap area. Ns = not significantly different; n=5 mice. Data are represented as mean ± SEM. (**D**) Number of terminals identified (count of terminal number) in size-matched analysis regions either within or outside the GPi overlap zone. *p=0.0317 by Mann-Whitney *U*-test; n=5 mice. Data are represented as mean ± SEM.

The online version of this article includes the following source data for figure 7:

**Source data 1.** Terminal size and count values for each individual animal, as displayed in *Figure 7C and D*.

---

from terminals of a wide range of sizes, including smaller ones that overlap with modulators (*Petrof and Sherman, 2013*; *Prasad et al., 2020*). Furthermore, while our analysis of synaptic terminal size confirmed sparser Cb innervation of the GPi overlap zone (*Deniau et al., 1992*), we were unable to detect a difference in bouton size between the GPi overlap and non-overlap zones. Collectively, all our data are consistent with Cb inputs being homogeneously drivers. We thus suggest that $M1_{L5}$ and Cb inputs to motor thalamus transmit relevant information to be relayed to cortex; though, we also recognize that the GABAergic inputs from GPi and/or SNr are capable of blocking these strong excitatory drives, making future functional studies critical.

## Beyond the classical notion of a cortico-basal ganglia-thalamocortical loop for motor control

Our experiments reveal three related insights regarding the relationship between cortex, basal ganglia, and motor thalamus that have important functional implications. We speculate on these below.

First, the textbook view of the basal ganglia situates it within an information loop from cortex to basal ganglia to thalamus and back to cortex (*Kandel et al., 2020*), though updates have been made (*Haber and Mcfarland, 2001*; *Lanciego et al., 2012*; *Goldberg et al., 2013*). However, our finding

that basal ganglia inputs to motor thalamus impinge on relays receiving excitatory, driving input from either M1$_{L5}$ or Cb pathways suggests a previously untested function of the basal ganglia—to gate information flow through motor thalamus.

Motor and frontal cortices are a major source of inputs to, as well as primary targets of, motor thalamic relays (*Bosch-Bouju et al., 2013*). These thalamocortical connections can be reciprocal (*Bosch-Bouju et al., 2013*; *Collins et al., 2018*; *Guo et al., 2018*) or non-reciprocal (*Rouiller et al., 1998*; *McFarland and Haber, 2002*). Indeed, here we directly demonstrate the presence of a feed-forward transthalamic pathway from M1, through VA/VL, to M2 (*Figure 1*). Therefore, in areas of the motor thalamus receiving convergent input from cortical L5 and basal ganglia, some fraction of which participates in transthalamic pathways, it is plausible that the basal ganglia can gate excitatory information between cortical regions (*Figure 8A*). When basal ganglia outputs are active, specific thalamic relays involved in transthalamic information transfer are inhibited. Since cortical areas have both direct and transthalamic connections organized in parallel (*Sherman and Guillery, 2013*; *Sherman, 2022*), disconnecting the transthalamic excitatory stream allows only the direct, corticocortical pathway to persist. In contrast, when thalamic relays are disinhibited due to upstream inhibition of the GPi or SNr, transthalamic transmission flows (*Figure 8A*). This would also apply to reciprocal cortico-thalamo-cortical connections (i.e. cortico-thalamocortical loops), which are well represented in regions of motor thalamus (*Guo et al., 2020*; *Guo et al., 2018*). Thus, we reason that a potential function of basal ganglia inputs to some fraction of motor thalamic relays is to gate L5 signals, and thereby affect communication between cortical areas (*Figure 8A*). Note that this offers a plausible and more detailed neuronal substrate for various ideas that the basal ganglia are involved in action selection (*Redgrave et al., 1999*; *Redgrave et al., 2011*; *Gaidica et al., 2018*; *Park et al., 2020*; *Fiore et al., 2021*). However, we recognize additional experiments in intact animals are required to test this speculation.

Second, previous anatomical data suggest a near-complete segregation of inputs from the Cb and basal ganglia to the motor thalamus (*Anderson and DeVito, 1987*; *Deniau et al., 1992*; *Sakai et al., 1996*; *Kuramoto et al., 2011*) and, while our experiments largely confirm this organization (*Figures 2 and 4*), we also demonstrate that these pathways in fact converge at the single cell level in a subset of motor thalamus. This finding extends prior transsynaptic labeling studies in non-human primates that highlighted an indirect interface between these systems (*Hoshi et al., 2005*; *Bostan et al., 2010*) by revealing a direct substrate for their interaction in mice. Interestingly, we find that anatomical overlap in VA/VL between the GPi and Cb inputs is greater than those of SNr and Cb, whereas the overlap shows the reverse for M1$_{L5}$ inputs, which overlap more with SNr than with GPi inputs. This suggests potential specialization in the gating functions served by the distinct basal ganglia inputs: perhaps SNr is specialized to gate cortical inputs, while GPi is specialized to gate those from Cb.

Third, our data advance understanding of basal ganglia function by demonstrating a direct projection from L5 of cortex, the starting point of the classical basal ganglia loop (*Figure 8B*, left), to the endpoint of the loop—motor thalamic relays. In other words, some M1$_{L5}$ axonal branches bypass the basal ganglia entirely (*Figure 8B*, right), which extends previous neuroanatomical data across species (*Mooney and Konishi, 1991*; *Farries et al., 2005*; *Kita and Kita, 2012*). The important aspect here is that absent this evidence, one might have concluded that M1$_{L5}$ innervates a discrete subset of motor thalamic relays that do not receive a basal ganglia input, thereby preserving the traditional cortico-basal ganglia-thalamocortical loop. Together with the findings that most, if not all, thalamic-projecting M1$_{L5}$ neurons branch to innervate the caudoputamen (*Economo et al., 2018*; *Jiang et al., 2020*), the intriguing, albeit untested, possibility arises that a single M1$_{L5}$ corticofugal neuron can affect motor thalamic activity indirectly through the classical basal ganglia loop (which is disinhibitory) as well as through the direct projection (excitatory). Such an organization has been anticipated in songbirds (*Goldberg and Fee, 2012*; *Goldberg et al., 2013*). Still, the precise organization of these branches (i.e. whether an M1$_{L5}$ neuron can target an individual thalamic relay through both pathways) is unknown. Even more, how signals organized in this fashion ultimately contribute to motor function requires much further experimentation.

Clear next steps are: (1) to understand how motor thalamus integrates these various inputs is through causal behavioral experiments and (2) to investigate whether the inhibitory gating of trans-thalamic pathways occurs across higher order thalamic nuclei.

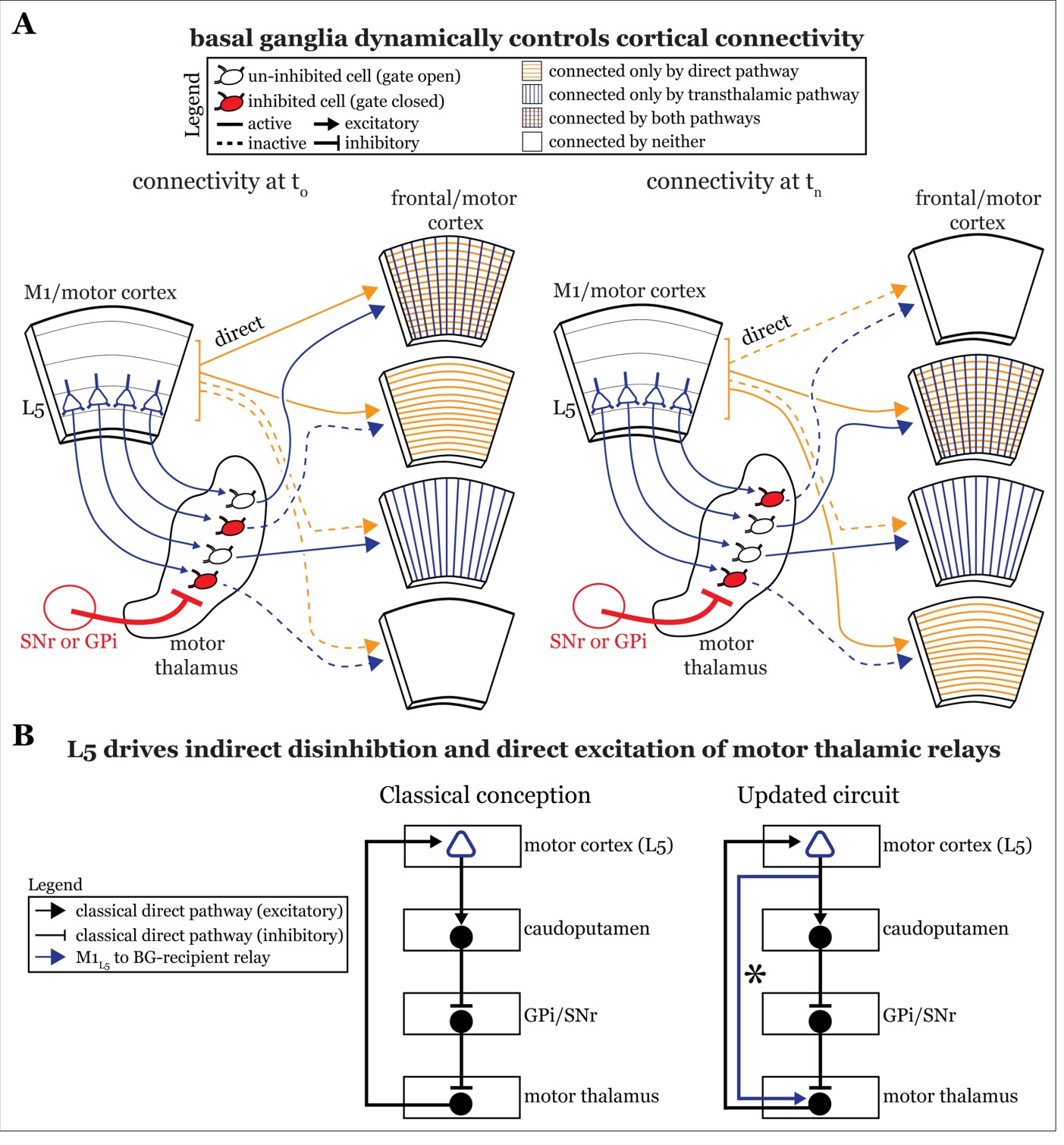

**Figure 8.** The basal ganglia gate information flow between cortical regions. (**A**) Schematic of the circuit organization of motor thalamus with respect to cortex and basal ganglia, showing the direct (orange) and transthalamic (blue) pathways through motor thalamus. When basal ganglia projections (red) to motor thalamic relays are active, they potently inhibit (red filled motor thalamic cells) and thereby *gate* transthalamic information flow between cortical regions. In contrast, when basal ganglia inputs to motor thalamus are themselves inhibited (open fill motor thalamic cells), they allow transthalamic information flow between cortical regions, where it may encounter signals from the direct cortico-cortical pathways. Via this gating function, the basal ganglia can regulate which cortical areas are connected by only direct corticocortical pathways, only transthalamic pathways, both, or neither. Importantly, this regulation is dynamic, such that the pattern of connected areas can change over short time scales (see *left*, time $T_0$, versus *right*, time $T_n$). (**B**) Box diagram of the basal ganglia circuitry according to the traditional textbook view (*left*) and with the novel direct connection between

*Figure 8 continued on next page*

*Figure 8 continued*

motor cortex layer 5 (L5) and motor thalamus shown (*right*; asterisk). That is, our data demonstrate that L5 of primary motor cortex (M1$_{L5}$) drivers innervate cells receiving internal segment of the globus pallidus (GPi) and substantia nigra pars reticulata (SNr) inputs. Given this organization, M1$_{L5}$ neurons could simultaneously disinhibit (via the basal ganglia loop, black arrows) and drive (via the direct corticothalamic projection, blue arrow) motor thalamic relays.

## Materials and methods

### Animals

Experiments were approved by the Institutional Animal Care and Use Committee at the University of Chicago under protocol #72313. Transgenic mice expressing cre recombinase in L5 of cortex were bred by crossing male Rbp4-cre KL100Gsat/Mmcd mice (GENSAT RP24-285K21; MGI:4367067) with female C57BL/6J mice. Cre-positive offspring of both sexes were used in experiments.

### Stereotaxic injections for optogenetics and slice physiology

Stereotactic injections of virus were performed as previously described (*Economo et al., 2018*; *Miller-Hansen and Sherman, 2022*) using either a 0.5 µl or 1 µl Hamilton syringe at postnatal days 24–28. Use of AAVs was approved by the Institutional Biosafety Committee protocol #1138. For tricolor fluorescence imaging experiments, 80 nl of AAV5-hSyn-hChR2(H134R)-EYFP (UNC Vector core) was injected into the GPi, 300 nl of AAV5-hSyn-hChR2(H134R)-mCherry (UNC Vector core) was injected into the deep Cb, and 400 nl of AAV8-Ef1a-Con/Foff 2.0-BFP (Addgene viral prep #137130-AAV8) was injected into M1. For optogenetics experiments, 400 nl (50 nl/min injection rate) of either AAV5-EF1a-double floxed-hChR2(H134R)-mCherry-WPRE-HGHpA (Addgene viral prep #20297-AAV5) or AAV5-EF1a-double floxed-hChR2(H134R)-EYFP-WPRE-HGHp (item #20298-AAV5) was injected into right M1 (depending on the fluorophore used for GPi and SNr injections) and 80–100 nl of either AAV5-hSyn-hChR2(H134R)-EYFP (UNC Vector core) or pAAV9-mDlx-ChR2-mCherry-Fishell-3 (Addgene viral prep #83898) was injected into the right GPi or SNr. Alternatively, in experiments analyzing Cb inputs, 300 nl of AAV-hSyn-hChR2(H134R)-mCherry (UNC Vector Core) was injected into left Cb. Only two inputs were injected for a given experiment, according to the description in the figures (e.g. Cb and GPi). Two weeks were given for expression before recordings were obtained.

The procedure used for injections of the retrograde label, fluororuby (5% in sterile saline; catalog: D1817, Thermo Fisher) was entirely the same, with the exception that fluororuby injections (300 nl) were placed in M2 (*Wang et al., 2020*; *Terada et al., 2022*) and were performed 4–7 days before the day of recording. Our injection coordinates were biased toward the rostral segment M2 to avoid leakage of dye near the M1 injection site (whereas some studies center M2 around +1.5 AP from bregma; *Jeong et al., 2016*).

The stereotactic coordinates used were: right M1: –1.6 ML, +1.5 AP, –0.6 DV from dura surface; right GPi: with a mediolateral angle of 6°, –2.7 ML, –0.7 AP, –4.0 DV; left Cb: +1.3 ML, –5.9 AP, –3.1 DV; right SNr: –1.3 ML, –3.1 AP, –4.3 DV; right M2: –1.0 ML, +2.2 AP, –0.5 DV.

### Tissue preparation for histology and fluorescence microscopy

As described previously (*Miller-Hansen and Sherman, 2022*) animals were transcardially perfused with phosphate-buffered saline followed by 4% paraformaldehyde in phosphate-buffered saline, pH 7.4. The brain was extracted and postfixed in 4% paraformaldehyde overnight at 4°C before transferring to a cold 30% sucrose solution for >48 hr. Brains were then cryosectioned coronally at 50 µm thickness on a sliding microtome.

Brain sections were mounted on Superfrost Plus (Fisher Scientific) slides and coverslipped with Vectashield (Vector Laboratories). A microscope with a 100 W mercury lamp with fluorescence optics (Leica Microsystems) was used to image the sections and photos were taken with a Retiga 2000 monochrome CCD camera and Q Capture Pro software (QImaging). Fiji (NIH) was used to overlay images.

For analysis of synaptic terminal size, high-resolution (1024×1024, 16-bit) z-stacks (10–15 z-planes per image) were taken using an LSM710 confocal microscope (Zeiss) at a z-plane interval of 5 µm or 10 µm.

## Tricolor labeling analysis

Fluorescence images for three mice with tricolor anterograde labeling (described above) were taken under identical conditions at two coronal planes (–0.7 AP and –1.0 AP) according to the Paxinos and Franklin's mouse brain atlas. Each channel, corresponding to either $M1_{L5}$ (blue), Cb (red), or GPi (green) for each animal, was analyzed separately. To average the fluorescence across animals in order to achieve a semi-quantitative assessment of terminal zones, the fluorescence values for each pixel within a rectangular region of interest (ROI) that encapsulated the motor thalamus (as well as the area immediately surrounding it) was extracted for each channel. Then, the background fluorescence value (a region of the tissue containing no terminal fluorescence) was subtracted from all pixels. The background-subtracted raw fluorescence values were then averaged across the three mice. An image of the ROI was then reconstructed from the raw fluorescence values by utilizing the 'text image' import function in Fiji. A binary mask of the pixels containing terminal fluorescence was generated by thresholding the image at two thresholds: 10% (more stringent) and 20% (less stringent) to generate the input maps for dense and sparse terminal labeling, respectively. This was repeated for each label/channel and overlayed onto the coronal sections displayed in *Figure 2* by aligning the slices from which the fluorescence values were taken to the atlas.

Analysis of the overlap between inputs (*Figures 2D and 4D*) was performed using images processed via the above method, using the less stringent threshold (examples shown in *Figures 2D and 3D*). The thresholded images were colored green (GPi or SNr), red (Cb), and blue ($M1_{L5}$) and compiled into a composite image, for which pixels receiving multiple inputs were distinguished with a unique color. For instance, pixels that were positive for both GPi (green) and Cb (red) inputs were colored yellow (green + red = yellow). Pixels of each color within an ROI encompassing VA/VL of the motor thalamus (i.e. excluding VM) were then counted using the 'color counter' plugin in Fiji. The percentage of overlapping pixels was then normalized to the sum of pixels from both inputs (e.g. the number of GPi/Cb overlapped [yellow] pixels was divided by the sum of the number of GPi and Cb pixels). This gives a percentage overlap that considers the volume of the individual projections, which is quantified for each set of inputs and displayed in *Figures 2D and 4D*.

## Acute slice preparation and whole-cell recordings

Animals were deeply anesthetized (nonresponsive to toe pinch) and immediately transcardially perfused with 8 ml of ice-cold oxygenated (95% $O_2$, 5% $CO_2$) artificial cerebrospinal fluid, which contained the following (in mM): 125 NaCl, 25 $NaHCO_3$, 3 KCl, 1.25 $NaH_2PO_4$, 1 $MgCl_2$, 2 $CaCl_2$, a 25 glucose. The brain was extracted, glue-mounted on a vibratome platform (Leica) for either standard coronal slices, at a 30° AP angle from the coronal plan, or at a 55° ML angle from the coronal plane to preserve descending cortical L5 axons (*Agmon and Connors, 1991*) for analysis of Cb or L5 synaptic properties, respectively (*Jackman et al., 2014*; *Economo et al., 2018*), and sliced in the same solution (ice-cold). Slices were cut at 385 µm thickness. Brain slices were then transferred to 33°C oxygenated artificial cerebrospinal fluid that was allowed to return to room temperature thereafter. This recovery in artificial cerebrospinal fluid occurred in the dark for 1 hr before recordings began.

Slices containing terminals from the described inputs were visualized using differential interference contrast with an Axioskop 2FS microscope (Carl Zeiss). Fluorescence from ChR2 expression was confirmed using the 5× air objective and guided recording locations. Recordings were made with a Multiclamp 700B amplifier and pCLAMP software (Molecular Devices). Recording glass pipettes with 4–6 MΩ resistance were filled with intracellular solution containing the following (in mM): 127 K-gluconate, 3 KCl, 1 $MgCl_2$, 0.07 $CaCl_2$, 10 HEPES, 0.1 EGTA, 2 $Na_2$-ATP, 0.3 Na-GTP, pH 7.3, 290 mOsm. Pharmacological inactivation of iGluRs was induced by bath application of 50 µM DNQX and 100 µM AP5 (Tocris). Blockade of mGluRs was attempted by bath application of 40 µM LY367385 and 30 µM MPEP (combined; Tocris). Pharmacological inactivation of $GABA_A$ receptors was performed by bath application of 20 µM SR 95531 (gabazine), while block of $GABA_B$ receptors was performed by application of 25 µM CGP 46381 (Tocris). The locations of each patched cell were logged and displayed in the figures.

Optogenetic stimulation was performed as previously (*Economo et al., 2018*; *Miller-Hansen and Sherman, 2022*). Briefly, stimulation was delivered using a 355 nm laser (DPSS: 3505-100), controlled with galvanometer mirrors (Cambridge Technology) focused on the slice through a 5× air objective using custom software in MATLAB (MathWorks). First, focal photostimulation of the terminals was

performed to test for the presence or absence of the inputs of interest. If the cell received only an excitatory input (i.e. either M1$_{L5}$ or Cb, but no basal ganglia), the synaptic properties of this input were tested by distal photostimulation of the axons (as described in Results section). Four pulses of 1 ms duration were delivered at a range of interstimulus intervals, including 100 ms (10 Hz), 50 ms (20 Hz), and 25 ms (40 Hz) during recordings. To test for the presence of mGluR responses, 40 Hz optogenetic stimulation (or, in some cases, 20 pulses of 1 ms delivered at 12 ms ISI, 83 Hz [*Miller-Hansen and Sherman, 2022*] was used but lack of mGluRs was equally conclusive) and responses were recorded in current clamp. In rare cases (6/177 neurons, like those used for representative traces), a single neuron underwent bath infusion of gabazine (during which 10 sweeps of laser stimulation were performed) followed by DNQX (+APV) to conclusively demonstrate the presence of both inputs. Gabazine infusion was acute (~2 min) to allow washout and recovery of the inhibitory signal before applying DNQX. The whole process was typically performed in roughly 40 min, yet the success rate was modest.

## Analysis of synaptic terminal size

Confocal z-stacks (10–15 z-planes per image at 5 μm interplane interval) from five animals injected as above for Cb and GPi were analyzed for terminal size within and just surrounding the overlap zone. Images were acquired with roughly half of the view field containing the GPi terminal zone, while the other half encompassed the surrounding region that contained Cb terminals/fibers. The z-stacks were collapsed in a maximum intensity projection (Fiji) and ROIs were manually drawn that either encapsulated the GPi terminal zone or the surrounding area with care to ensure that the ROIs were matched in terms of their areas. Images were then segmented using the 'triangle' auto threshold algorithm in Fiji. Next, the pixels that satisfied the threshold for being counted, i.e., the synaptic terminals, were analyzed separately within each ROI using the 'analyze particles' tool in Fiji. The only specific parameter used is that a particle had to represent at least three pixels to be counted. This same analysis was performed for two z-stacks (one from each tissue section) for each animal, corresponding to the locations we recorded in (AP –0.7 and AP –1.0). The data for the two images were averaged to get the value for each animal.

## Data analysis and statistics

Electrophysiological data were collected using custom MATLAB software and analyzed using GraphPad Prism (v7.0). The amplitude of responses to stimulation pulses was measured by subtracting the average value for 20 ms before the delivery of a pulse (baseline) from the maximum value of the peak. The PPR was calculated by dividing the amplitude of the second pulse by that of the first pulse. Statistical tests involving two groups were performed using non-parametric Mann-Whitney *U*-tests, where indicated, in Prism. Image analysis was conducted in Fiji (NIH) and figures were produced using Adobe Illustrator.

## Acknowledgements

This work was supported by NIH grant NS094184.

## Additional information

### Funding

| Funder | Grant reference number | Author |
| --- | --- | --- |
| National Institute of Neurological Disorders and Stroke | NS094184 | S Murray Sherman |

The funders had no role in study design, data collection and interpretation, or the decision to submit the work for publication.

### Author contributions

Kevin P Koster, Conceptualization, Data curation, Formal analysis, Investigation, Visualization, Methodology, Writing – original draft, Writing – review and editing; S Murray Sherman, Conceptualization,

Resources, Supervision, Funding acquisition, Writing – original draft, Project administration, Writing – review and editing

### Author ORCIDs
Kevin P Koster  http://orcid.org/0000-0003-2935-3427
S Murray Sherman  http://orcid.org/0000-0002-1520-2778

### Ethics
All experiments were performed in accordance with protocols approved by the Institutional Animal Care and Use Committee at the University of Chicago.

### Decision letter and Author response
Decision letter https://doi.org/10.7554/eLife.97489.sa1
Author response https://doi.org/10.7554/eLife.97489.sa2

---

## Additional files

### Supplementary files
• MDAR checklist

### Data availability
All data generated or analyzed during this study are represented in the Figures. Mean values and sample variability are listed in the Results section of the text where applicable. Individual data points are represented on the graphs/Figures. In addition, source data files for all quantitative data are provided.

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
