## [Editor Report]

This study provides fundamental findings that challenge the traditional view of segregated cerebellar and basal ganglia circuits in the motor thalamus, revealing a novel intersection where GABAergic inputs from the basal ganglia converge with excitatory inputs from both the primary motor cortex and deep cerebellar nuclei. The evidence supporting these claims is compelling, utilizing rigorous optogenetic approaches and tricolor viral labeling to uncover the intricate circuit organization. These findings have theoretical and practical implications beyond a single subfield, advancing our understanding of how the brain processes motor commands.

---

## [Decision Letter]

**Decision letter after peer review:**

Thank you for submitting your article "Convergence of inputs from the basal ganglia with layer 5 of motor cortex and cerebellum in mouse motor thalamus" for consideration by *eLife*. Your article has been reviewed by 2 peer reviewers, one of whom is a member of our Board of Reviewing Editors, and the evaluation has been overseen by John Huguenard as the Senior Editor.

The reviewers have discussed their reviews with one another, and the Reviewing Editor has drafted this to help you prepare a revised submission. The reviewers have opted to remain anonymous.

Essential revisions (for the authors):

The manuscript provides a detailed examination of the motor thalamus's synaptic architecture, highlighting the convergence of inputs from the motor cortex, cerebellum, and basal ganglia in a manner that deviates from previously accepted models of segregated input processing. Employing techniques such as anatomical tracing, optogenetics, and electrophysiology, the research uncovers a degree of interconnectivity within these synaptic networks that surpasses earlier understandings. The identification of driver-type synaptic properties from motor cortex and cerebellar inputs presents a compelling case for reevaluating the functional connectivity of the thalamus. Nonetheless, the absence of conclusive evidence for the coalescence of all three inputs on individual thalamic neurons suggests a need for cautious interpretation of the data and points to the necessity for further study.

For the manuscript to effectively convey its contributions and enhance its impact, the following revisions are essential:

1) Revised Interpretation of Synaptic Convergence: Amend the discussion of synaptic convergence to more accurately reflect the tentative nature of the findings. The manuscript should articulate that the results infer potential sites of convergence that merit additional research to establish definitively. An adjustment in the narrative might read: "The data indicate potential sites within the motor thalamus where M1L5, Cb, and basal ganglia inputs might converge, setting a course for future inquiries to validate triple convergence."

2) Clarified Functional Implications: Restrain the presentation of the integrative roles posited for M1L5 and Cb inputs within the motor thalamus. The narrative should affirm that the hypothesized driver roles of these inputs, as suggested by synaptic characteristics, await empirical confirmation regarding their actual effects on thalamic output.

3) Addressing discrepancies and providing context: The manuscript should reconcile the differences between the anatomical and functional findings, specifically regarding the overlap of SNr and Cb inputs. It should also provide a clearer explanation for observed synaptic patterns, including a discussion on bouton sizes relative to driver and modulator synapses. This discussion should be placed within the context of existing literature to offer alternative explanations and a comprehensive view of the findings.

By addressing these points, the authors will enhance the manuscript's contribution to our understanding of the synaptic organization of the motor thalamus.

*Reviewer #1 (Recommendations for the authors):*

The authors have undertaken a methodologically robust study integrating anatomical tracing, optogenetics, and electrophysiological recordings to explore the synaptic connectivity within the motor thalamus. However, there are several concerns that need to be addressed to strengthen the conclusions drawn from this research:

1. The current conclusions regarding the convergence of inputs from M1L5, Cb, and basal ganglia predominantly reflect findings from dual-area stimulation, and the manuscript asserts convergence onto the same motor thalamus neurons. The lack of electrophysiological evidence supporting triple convergence necessitates a more cautious interpretation. The authors are encouraged to revise the text to reflect that the data currently suggest potential sites of convergence and that these observations warrant further investigation. A revised statement might read, "Our findings imply potential convergent sites within the motor thalamus for M1L5, Cb, and basal ganglia inputs; however, additional studies are required to definitively demonstrate triple convergence." The authors are best positioned to rephrase these sections to both accurately reflect the evidence and maintain the integrity of their findings.

2. While the manuscript challenges the notion of segregated cerebellar and basal ganglia inputs, the functional implications of this integration are not conclusively demonstrated. It is recommended that the authors review the sections discussing these implications and adjust the text to highlight the preliminary nature of this conceptual map, possibly presenting it as a hypothesis for subsequent research, for example, as a conceptual framework for understanding potential integrative functions within the motor thalamus.

3. The manuscript proposes that both M1L5 and Cb inputs serve as drivers within the motor thalamus, a claim inferred from synaptic properties. However, this study does not present functional evidence of the effects of these inputs on motor thalamic output. It would be prudent for the authors to clarify that while the synaptic analysis suggests driver-type properties, the actual functional impact requires further investigation. The authors may wish to refine the language in this section to clarify that these are indicative findings requiring further functional validation.

4. The manuscript would benefit from a broader discussion of the results, including alternative explanations for the observed synaptic patterns. The authors should consider incorporating a section that addresses possible differential interpretations of the data, thus providing a comprehensive view of the findings within the context of existing literature.

The authors are to be commended for their rigorous approach and the contribution this study makes to the field. Addressing these concerns will undoubtedly enhance the manuscript's impact and provide a clearer understanding of the motor thalamus's synaptic organization.

*Reviewer #2 (Recommendations for the authors):*

Please rephrase the following sentence to improve clarity:

Therefore, we investigated whether a feedforward transthalamic pathway existed in the VA/VL portion of the motor thalamus by probing for optogenetic responses from L5 of primary motor cortex (M1L5) in VA/VL relays projecting to secondary motor cortex (M2)

Line 79-80: corticothalamic should be replaced with thalamocortical:

Moreover, the traditional view of the basal ganglia projection to motor thalamus situates it within a cortical-basal ganglia-corticothalamic loop

Line 108:

Please specify that this mouse line is a layer 5 specific line.

Line 178: Typo; 10/62 should be replaced by 10/42 to match the reported percentage!

Line 208-215: Typo; Figure 4 should be replaced by Figure 5!

How do the authors explain the discrepancy between the findings of the anatomical (Figure 4) and functional (Figure 5) experiments in terms of the absence and presence of overlapping SNr and Cb inputs in VA/VL, respectively? This should be addressed at the end of line 215.

Please consider adding a discussion topic on this point, and the functional implications of the difference in SNr vs. GPi overlap with cerebellar terminals.

Line 224: please provide a brief context for how paired-pulse paradigm can help with distinguishing driver and modulator synapses. This is to help with making the paper accessible to a broader group.

Line 248: same as above, please provide some context for the importance of studying the size of synaptic terminals and how it can inform us about driver/modulator roles. Also, what threshold defines small vs big terminals? Please address that too.

Is the paper arguing that Cb and GPi terminals are both small, despite being drivers? If so, why do they not follow the typical phenomenon of modulator synapses having smaller boutons? Please clarify this in lines 308-317. If the paper is arguing that the terminals are big, then please explain the discrepancy with Deniau study.

How do GPi/Cb bouton sizes compare with SNr terminals?

Figure 1:

Please spell out DIC in the figure caption.

---

## [Author Response]

Essential revisions (for the authors):The manuscript provides a detailed examination of the motor thalamus's synaptic architecture, highlighting the convergence of inputs from the motor cortex, cerebellum, and basal ganglia in a manner that deviates from previously accepted models of segregated input processing. Employing techniques such as anatomical tracing, optogenetics, and electrophysiology, the research uncovers a degree of interconnectivity within these synaptic networks that surpasses earlier understandings. The identification of driver-type synaptic properties from motor cortex and cerebellar inputs presents a compelling case for reevaluating the functional connectivity of the thalamus. Nonetheless, the absence of conclusive evidence for the coalescence of all three inputs on individual thalamic neurons suggests a need for cautious interpretation of the data and points to the necessity for further study.For the manuscript to effectively convey its contributions and enhance its impact, the following revisions are essential:1) Revised Interpretation of Synaptic Convergence: Amend the discussion of synaptic convergence to more accurately reflect the tentative nature of the findings. The manuscript should articulate that the results infer potential sites of convergence that merit additional research to establish definitively. An adjustment in the narrative might read: "The data indicate potential sites within the motor thalamus where M1L5, Cb, and basal ganglia inputs might converge, setting a course for future inquiries to validate triple convergence."

We have now clarified this point with several pieces of added text. We have clarified the abstract to separate discussion of M1L5 and Cb convergence with the basal ganglia. In addition, we have added a similar language to that suggested in the first paragraph of the discussion to better frame that we are only focused on convergence of any two inputs (excitatory and inhibitory). Though we agree future study using alternative methods may be crucial to an understanding of Cb and M1_L5_ convergence.

2) Clarified Functional Implications: Restrain the presentation of the integrative roles posited for M1L5 and Cb inputs within the motor thalamus. The narrative should affirm that the hypothesized driver roles of these inputs, as suggested by synaptic characteristics, await empirical confirmation regarding their actual effects on thalamic output.

With respect, we feel that the interpretation we offer is not unreasonable, although based on the comments, we have softened them and make clearer that these are plausible conclusions and speculations based on the actual data. The driver classification of glutamatergic inputs to thalamus is well established in the literature, and where identified, represents the main source of information to be relayed to cortex (e.g., retinal input to the LGN). Furthermore, transthalamic relays are now an accepted feature of cortical organization, and considerable data points to L5 inputs being a driver input to these circuits. Thus, we argue, it is a reasonable suggestion that a GABAergic input (from GPi or SNr) to relay cells also receiving a L5 input can act to gate the transthalamic circuit involved. The same logic applies to Cb inputs being driver and converging with basal ganglia GABAergic inputs. Still, we recognize that the basal ganglia strongly suppress motor thalamic relays at baseline, and can thereby block strong excitatory signals from these driving inputs. We highlight this distinction in the Discussion (lines 360-362).

3) Addressing discrepancies and providing context: The manuscript should reconcile the differences between the anatomical and functional findings, specifically regarding the overlap of SNr and Cb inputs. It should also provide a clearer explanation for observed synaptic patterns, including a discussion on bouton sizes relative to driver and modulator synapses. This discussion should be placed within the context of existing literature to offer alternative explanations and a comprehensive view of the findings.

We have addressed this apparent discrepancy between the anatomical and physiological data in the manuscript and have provided a brief explanation below as well (reviewer #2, point 7). Briefly, while the areas of dense terminal innervation of motor thalamus between Cb and SNr are very limited, there is sufficient overlap of sparse terminal fields to account for substantial functional convergence.

Further, we provide much more background on driver/modulator classification and terminal size significance in both the Results and Discussion sections (see lines 224-236; 327-334; 353-354).

Reviewer #1 (Recommendations for the authors):The authors have undertaken a methodologically robust study integrating anatomical tracing, optogenetics, and electrophysiological recordings to explore the synaptic connectivity within the motor thalamus. However, there are several concerns that need to be addressed to strengthen the conclusions drawn from this research:1. The current conclusions regarding the convergence of inputs from M1L5, Cb, and basal ganglia predominantly reflect findings from dual-area stimulation, and the manuscript asserts convergence onto the same motor thalamus neurons. The lack of electrophysiological evidence supporting triple convergence necessitates a more cautious interpretation. The authors are encouraged to revise the text to reflect that the data currently suggest potential sites of convergence and that these observations warrant further investigation. A revised statement might read, "Our findings imply potential convergent sites within the motor thalamus for M1L5, Cb, and basal ganglia inputs; however, additional studies are required to definitively demonstrate triple convergence." The authors are best positioned to rephrase these sections to both accurately reflect the evidence and maintain the integrity of their findings.

We are sorry about our lack of clarity on this point. We have amended the manuscript to prevent other readers from having the same interpretation (please see the response to point #1 above).

2. While the manuscript challenges the notion of segregated cerebellar and basal ganglia inputs, the functional implications of this integration are not conclusively demonstrated. It is recommended that the authors review the sections discussing these implications and adjust the text to highlight the preliminary nature of this conceptual map, possibly presenting it as a hypothesis for subsequent research, for example, as a conceptual framework for understanding potential integrative functions within the motor thalamus.

Thank you for this suggestion. Indeed, it was our intention to speculate on the potential implications of the convergence we show and not to suggest our interpretation as a definitive functional consequence. We have added some text just to highlight the speculative nature of much of the discussion and have added some text to specifically address this point (lines 360-362).

3. The manuscript proposes that both M1L5 and Cb inputs serve as drivers within the motor thalamus, a claim inferred from synaptic properties. However, this study does not present functional evidence of the effects of these inputs on motor thalamic output. It would be prudent for the authors to clarify that while the synaptic analysis suggests driver-type properties, the actual functional impact requires further investigation. The authors may wish to refine the language in this section to clarify that these are indicative findings requiring further functional validation.

See response to point #2 of “Essential revisions”.

4. The manuscript would benefit from a broader discussion of the results, including alternative explanations for the observed synaptic patterns. The authors should consider incorporating a section that addresses possible differential interpretations of the data, thus providing a comprehensive view of the findings within the context of existing literature.

We feel we have done a reasonable job of assessing the limitations of our study (experimental provisos section and elsewhere) but have amended the discussion at several points to highlight where the data ends and the speculation begins. We hope this is sufficient.

Reviewer #2 (Recommendations for the authors):Please rephrase the following sentence to improve clarity:Therefore, we investigated whether a feedforward transthalamic pathway existed in the VA/VL portion of the motor thalamus by probing for optogenetic responses from L5 of primary motor cortex (M1L5) in VA/VL relays projecting to secondary motor cortex (M2)

Clarified. See lines 104-108.

Line 79-80: corticothalamic should be replaced with thalamocortical:

Done. Thank you.

Moreover, the traditional view of the basal ganglia projection to motor thalamus situates it within a cortical-basal ganglia-corticothalamic loop

Clarified. See lines 80-83.

Line 108:Please specify that this mouse line is a layer 5 specific line.

Done. Thank you.

Line 178: Typo; 10/62 should be replaced by 10/42 to match the reported percentage!

Fixed. We are sorry about that.

Line 208-215: Typo; Figure 4 should be replaced by Figure 5!

Thank you. Fixed. Again, sorry about that egregious mistake.

How do the authors explain the discrepancy between the findings of the anatomical (Figure 4) and functional (Figure 5) experiments in terms of the absence and presence of overlapping SNr and Cb inputs in VA/VL, respectively? This should be addressed at the end of line 215.

This is a good question and we have now addressed it in the Results where you suggest (now lines 218-221). The short answer is that while the dense terminal fields in motor thalamus do not overlap much, there is enough overlap of sparser terminal zones to allow for a substantial proportion of cells receiving convergent input.

Please consider adding a discussion topic on this point, and the functional implications of the difference in SNr vs. GPi overlap with cerebellar terminals.

We have now added a short text piece on this point as noted above. We are to this point unsure of the significance of the difference in overlap between Cb and GPi, vs. Cb and SNr, but note this in the discussion (lines 406-411).

Line 224: please provide a brief context for how paired-pulse paradigm can help with distinguishing driver and modulator synapses. This is to help with making the paper accessible to a broader group.

We have provided much more background. Thank you for this suggestion. See lines 224-236.

Line 248: same as above, please provide some context for the importance of studying the size of synaptic terminals and how it can inform us about driver/modulator roles. Also, what threshold defines small vs big terminals? Please address that too.

Please see lines 224-236, 327-334, and 353-354.

Is the paper arguing that Cb and GPi terminals are both small, despite being drivers? If so, why do they not follow the typical phenomenon of modulator synapses having smaller boutons? Please clarify this in lines 308-317. If the paper is arguing that the terminals are big, then please explain the discrepancy with Deniau study.

Sorry about the confusion here. We mean to say that the glutamatergic inputs from the Cb appeared to have a different abundance and terminal size depending on whether they were in the basal ganglia overlap zone or in the Cb-only region of motor thalamus. This is the notion from the from Deniau study, which was qualitative. However, we were imprecise in our original presentation. Deniau observed smaller Cb terminals in VM motor thalamus, whereas we focus here on VA/VL. We have edited the text to include this point and made other clarifications (see 271-287, especially 285-287).

How do GPi/Cb bouton sizes compare with SNr terminals?

In this study we did not quantify the terminal sizes of the GABAergic projections from GPi or SNr and think that it is largely unnecessary, since there is no known functional correlate like there is for glutamatergic terminals. However, we have clarified our discussion of Cb terminal sizes in the points raised above.

Figure 1:Please spell out DIC in the figure caption.

Done.